# AN ASYMPTOTIC THEORY OF RANDOM SEARCH FOR HYPERPARAMETERS IN DEEP LEARNING

## ABSTRACT

Scale is essential in modern deep learning; however, greater scale brings a greater need to make experiments efficient. Often, most of the effort is spent finding good hyperparameters, so we should consider exactly how much to spend searching for them—unfortunately this requires a better understanding of hyperparameter search, and how it converges, than we currently have. An emerging approach to such questions is *the tuning curve*, or the test score as a function of tuning effort. In theory, the tuning curve predicts how the score will increase as search continues; in practice, current estimators use nonparametric assumptions that, while robust, can not extrapolate beyond the current search step. Such extrapolation requires stronger assumptions—realistic assumptions designed for hyperparameter tuning. Thus, we derive an asymptotic theory of random search. Its central result is a new limit theorem that explains random search in terms of four interpretable quantities: the effective number of hyperparameters, the variance due to random seeds, the concentration of probability around the optimum, and the best hyperparameters' performance. These four quantities parametrize a new probability distribution, *the noisy quadratic*, which characterizes the behavior of random search. We test our theory against three practical deep learning scenarios, including pretraining in vision and fine-tuning in language. Based on 1,024 iterations of search in each, we confirm our theory achieves excellent fit. Using the theory, we construct the first confidence bands that extrapolate the tuning curve. Moreover, once fitted, each parameter of the noisy quadratic answers an important question—such as what is the best possible performance. So others may use these tools in their research, we make them available at (URL redacted).

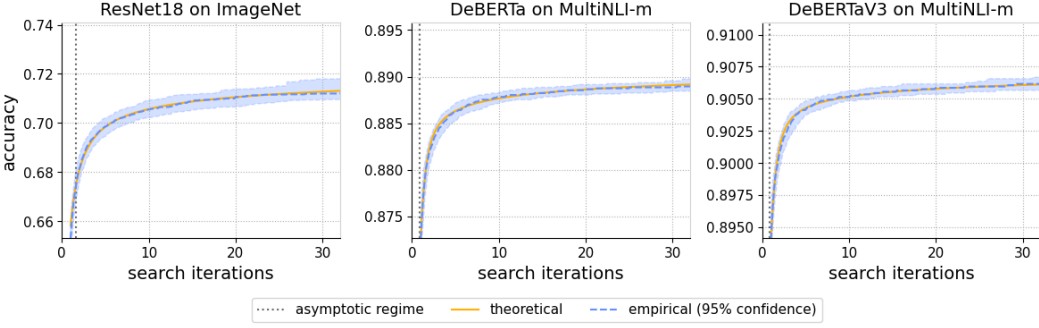

Figure 1: Our asymptotic theory predicts functional forms with excellent fit to the ground truth. Our theory explains how performance improves with increased tuning effort—a trade-off captured by *the tuning curve*: the validation score as a function of the number of search iterations. Each plot compares the *empirical* tuning curve against its *theoretical* form using 1,024 iterations of search. *Across three models, two datasets, and both vision pretraining and language fine-tuning, the ground truth curve closely adheres to the theoretical form which remains fully in the 95% confidence bands. Better yet, while the results are asymptotic in theory, in practice they can apply after 1 or 2 iterations.*

# 1 INTRODUCTION

Deep learning advances through experimentation, but as models have grown in scale, experiments have grown in cost. As cost becomes the bottleneck, researchers must compromise either rigor or speed—unless experiments become more efficient. In many experiments, the most expensive step is finding good hyperparameters; thus, we should only spend the necessary effort to search for them. The issue is: this search is often a blackbox, with few tools for understanding it or how it converges.

To answer such questions, an emerging approach is to estimate the *tuning curve*, or the test score as a function of tuning effort (Dodge et al., 2019; Lourie et al., 2024). Figure 1 illustrates an example. The $x$-axis measures tuning effort (e.g., iterations, compute), and the $y$-axis measures performance (e.g., F1, perplexity). The tuning curve reveals where the score levels off and what the best possible score might be, removing many subjective judgments previously required in experiments. However, while the tuning curve might clarify convergence, its estimators are nonparametric and thus do not extrapolate. What is more: the tuning curve *describes* search, but does not *explain* it. If a model tunes slowly, is the search space too big or the problem just difficult? Predicting future progress, explaining how search progresses, calls for something of greater strength.

Thus, we derive an asymptotic theory of random search. Its central result is a novel limit theorem characterizing the tail of random search. Focusing on the better scores, their distribution converges to a new family: *the noisy quadratic distribution*. This family explains random search in terms of four interpretable quantities: the effective number of hyperparameters, the variance due to random seeds, the concentration of probability around the optimum, and the best hyperparameters' performance. The theory is mechanistic—deriving from how random search works—and built on two empirical precepts: the hyperparameter loss is smooth, and the noise when retraining is normal with constant variance. Remarkably, this simple structure emerges as you approach the optimal hyperparameters, suggesting a new discovery: *the asymptotic regime*. Empirically, the asymptotic regime governs random search after only a few iterations, thus the theory explains much of its behavior in practice.

Still, the ultimate test of any theory is how well it reflects the data. We validate our theory in three practical deep learning scenarios, including pretraining in vision and fine-tuning in language (§4). With 1,024 iterations in each, we assess how well our theoretical form fits the empirical distribution from random search (§4.1). In all three scenarios, the theoretical form adheres closely to the ground truth and remains within its 95% confidence bands. Beyond fit, we test whether the noise is actually normal with constant variance (§4.2). In fact, it converges to normality long before the asymptotic regime, and while the variance begins inflated it quickly converges to a constant. Last, we test our theory's application. In each scenario, we construct point estimates and confidence bands using 48 search iterations (§4.3). The point estimates mostly smooth their nonparametric baselines; however, the confidence bands show a dramatic improvement. While the nonparametric bands become trivial after a fraction of the total iterations, the parametric confidence bands extrapolate beyond them.

Our theory reflects the data, derives from realistic assumptions, and extrapolates the tuning curve with complete statistical rigor; therefore, it provides a solid foundation for experiments involving hyperparameters. Beyond tuning curves, it offers other tools for researchers and practitioners to better understand their models. For example, each parameter of the noisy quadratic answers its own question, such as: what is the *effective* number of hyperparameters? How much variation is due to random seeds? Or, what is the best possible performance? Using standard statistical techniques, each of these parameters can be estimated with confidence. So that others may use these tools in their research, we make them available at (URL redacted).

---

**A Limit Theorem for Random Search**

Theoretically under regularity conditions & empirically for deep learning:

**Minimization.** When *minimizing* via random search, the *left* tail of the score distribution converges to a *convex* noisy quadratic distribution:

$$\mathcal{Q}_{\min}(\alpha, \beta, \gamma, \sigma)$$

**Maximization.** When *maximizing* via random search, the *right* tail of the score distribution converges to a *concave* noisy quadratic distribution.

$$\mathcal{Q}_{\max}(\alpha, \beta, \gamma, \sigma)$$

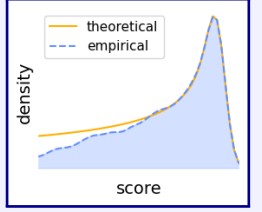

*Example of Maximization*

## 2 An Asymptotic Theory of Random Search

Let us develop the asymptotic theory of random search.[1] We begin by describing the formalism, then present the theory in two parts. First, we assume the score is a deterministic function of the hyperparameters. This assumption permits straightforward analysis; however, it is too simple to describe most applications of interest. Thus, we extend this analysis with the additive noise from random elements such as the initialization and data ordering. Empirical results show this approach offers an excellent model of random search in deep learning (§4).

### 2.1 Formalizing Random Search

Imagine fitting a neural network—perhaps you pretrain a ResNet on image classification.[2] Many choices must be made: Which architecture? What learning rate? How much regularization? Each choice is a hyperparameter, and each hyperparameter takes a value from the *hyperparameter search space*, $\boldsymbol{x} = [x_1, \ldots, x_d] \in \mathbb{X}$. Fixing these values and evaluating the network produces a score, $y \in \mathbb{Y} \subset \mathbb{R}$, such as accuracy, that we wish to optimize.

Optimizing this score means optimizing the hyperparameters; and for that, we use a hyperparameter tuning algorithm. Such algorithms typically sample choices from a stochastic policy, so we treat both the choices, $\boldsymbol{X}_1, \ldots, \boldsymbol{X}_n$, and their scores, $Y_1, \ldots, Y_n$, as random variables. Since sampling the next choice costs only a fraction of evaluating it, we use the number of evaluations, $n$, to measure the cost. Of course, the cost of an evaluation can vary greatly between models and search spaces, so each evaluation should be normalized by its average cost when making such comparisons.

We capture hyperparameter tuning's progress as a function of cost by the *tuning process*:

$$T_k := \max_{i=1\ldots k} Y_i \tag{1}$$

In general, the tuning process depends on both the model and the hyperparameter tuning algorithm.

While one could use any algorithm to optimize hyperparameters for *deployment*, research requires a standard pick to ensure fair comparisons in *development*. One simple, robust, and surprisingly effective standard is random search. *Random search* independently draws choices from the *search distribution*, $\boldsymbol{X}_i \sim \mathcal{X}$. Evaluating a choice then yields a score from *the score distribution*, $Y_i \sim \mathcal{Y}$.

Under random search, analyzing the tuning process is particularly tractable because the choices, $X_i$, and thus the scores, $Y_i$, are independent and identically distributed (i.i.d.). Intuitively, since all scores are i.i.d., the best score after $k$ rounds is just the maximum from a sample of size $k$. As a result, the CDF of $Y_i$, $F(y) = P(Y_i \leq y)$, and that of $T_k$, $F_k(y) = P(T_k \leq y)$, share a relationship:

$$F_k(y) = P\left(\max_{i=1\ldots k} Y_i \leq y\right) = \prod_{i=1\ldots k} P(Y_i \leq y) = F(y)^k \tag{2}$$

Essentially, the distribution from one round of random search defines the distribution from $k$ rounds of random search. Then after $n$ rounds, we have a sample of size $n$ with which to estimate that distribution. Using these insights, we can estimate properties of the entire tuning process.

Still, the tuning process is a complex joint distribution that is difficult to interpret or compare. Often, it is more convenient to consider a summary, such as the median. Following Lourie et al. (2024), we define $T_k$ ($k \in \mathbb{R}$) as the random variable with CDF $F(y)^k$. Then, letting $\mathbb{M}[X]$ denote the median of $X$, the *tuning curve* is the function, $\tau : \mathbb{R} \to \mathbb{Y}$:

$$\tau(k) := \mathbb{M}[T_k] \tag{3}$$

More generally, one might distinguish the *median*, $\tau_m(k) := \tau(k)$, and *expected*, $\tau_e(k)$ tuning curve:

$$\tau_e(k) := \mathbb{E}[T_k] \tag{4}$$

The tuning curve answers many questions a researcher might ask during model development. To find the best achievable performance, researchers can look at the tuning curve's limit. To check if a model is undertuned, researchers can see how performance increases with a few more iterations of search. To compare models while accounting for tuning effort, they can compare the tuning curves at various budgets, adjusting each curve by the average cost to train that model.

---

[1]We state the theory for maximization, minimization being equivalent. See §A and B for more formulas.

[2]Our formalization closely follows that of Lourie et al. (2024) (§3.1).

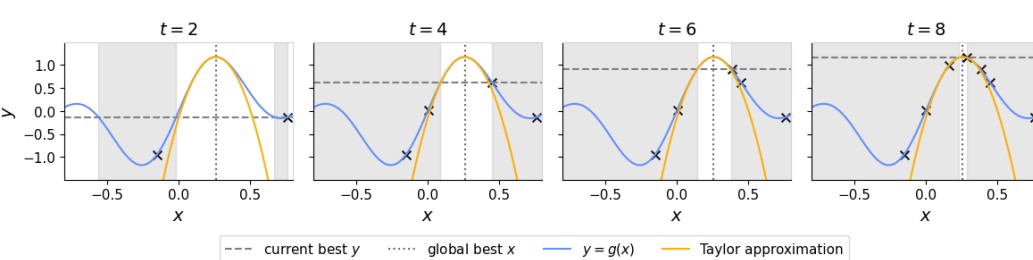

Figure 2: An illustration of random search. As search continues, the current best score increases and the region of better hyperparameters shrinks. As this region closes around the optimum, the Taylor polynomial offers a better and better approximation.

## 2.2 THE DETERMINISTIC CASE

Our theory starts with a simple intuition: at any time in random search, the only hyperparameters that matter are those better than the best you have seen so far. As search continuous, you find better hyperparameters, and the region of even better ones converges about the optimum. In this region, the Taylor polynomial becomes a better and better approximation to the underlying objective. Where the approximation is sufficiently good, we call *the asymptotic regime*. Figure 2 illustrates this idea.

At the optimum, $(\boldsymbol{x}^*, y^*)$, the Taylor series is dominated by the Hessian, $H_{\boldsymbol{x}^*}$, as the gradient is 0:

$$g(\boldsymbol{x}) \approx y^* + \frac{1}{2}(\boldsymbol{x} - \boldsymbol{x}^*)^T H_{\boldsymbol{x}^*}(\boldsymbol{x} - \boldsymbol{x}^*) \tag{5}$$

At the same time, any continuous probability density is roughly constant on a small enough interval; so, near the optimum, the search distribution is approximately uniform.

Putting these facts together, we derive the limit of the score distribution via a geometric argument. Consider the event $Y = g(\boldsymbol{X}) > y$. Rearranging the Taylor approximation, we obtain:

$$-\frac{1}{2}(\boldsymbol{x} - \boldsymbol{x}^*)^T H_{\boldsymbol{x}^*}(\boldsymbol{x} - \boldsymbol{x}^*) \leq y^* - y \tag{6}$$

Since $\boldsymbol{x}^*$ is a maximum, the Hessian is negative semi-definite, so this equation defines an ellipsoid. $\mathbb{P}(Y > y)$ is then proportional to the volume of this ellipsoid, which is proportional to $(y^* - y)^{d^*/2}$, where $d^*$ is the rank of the Hessian. Consequently, as $y \to y^*$ the CDF approximately satisfies:

$$1 - F(y) = \mathbb{P}(Y > y) \propto (y^* - y)^{d^*/2} \tag{7}$$

Motivated by this analysis, we define the *concave quadratic distribution*, $\mathcal{Q}_{\max}(\omega, \beta, \gamma)$, by:

$$F(y; \omega, \beta, \gamma) := 1 - \omega (\beta - y)^{\gamma/2} \tag{8}$$

Usually, we prefer an alternative, more interpretable parametrization. Let $\alpha$ be the minimum of the distribution's support. Then $F(\alpha) = 1 - \omega(\beta - \alpha)^{\gamma/2} = 0$, thus $\omega = (\beta - \alpha)^{-\gamma/2}$. So, for the CDF:

$$F(y; \alpha, \beta, \gamma) := 1 - \left(\frac{\beta - y}{\beta - \alpha}\right)^{\gamma/2} \tag{9}$$

Now, we can differentiate the CDF to obtain the PDF:

$$f(y; \alpha, \beta, \gamma) = \frac{\gamma}{2(\beta - \alpha)} \left(\frac{\beta - y}{\beta - \alpha}\right)^{\frac{\gamma - 2}{2}} \tag{10}$$

Each parameter has a nice interpretation in terms of the hyperparameter tuning problem: $\alpha$ measures how *concentrated* the distribution is near the maximum, $\beta$ is *the highest achievable score*, and $\gamma$ is *the effective number of hyperparameters*—which is always less than the nominal number.

To summarize, we introduce a new parametric family: *the quadratic distribution*. When maximizing via random search, the score distribution's *right* tail approaches the *concave* quadratic distribution; and, while we do not discuss it here, when minimizing the *left* tail approaches a similar limit, the *convex* quadratic distribution, which we give formulas for in §A.

## 2.3 THE STOCHASTIC CASE

So far, our theory assumes the score is deterministic given the hyperparameters; however, this is often not the case. More commonly, the score varies—even with the same hyperparameters—due to random factors such as the initialization, data order, or nondeterminism of the GPU. So, how can we incorporate such nondeterminism into our theory?

All else equal, we prefer the simplest approach; let us begin there, and only add complexity as necessary to obtain an accurate model. One thought is: apply our current theory to the conditional mean, $\mathbb{E}[Y|\boldsymbol{X}]$; $Y$ may be random, but its expectation is not. The problem is, we never observe the mean and in practice we keep models not hyperparameters so really $Y$ is the score we get.

Instead, let us take inspiration from classic regression analysis. If the mean varies according to our theory, perhaps $Y$ varies with additive noise about that mean? Formally, let $g(\boldsymbol{X}) = \mathbb{E}[Y|\boldsymbol{X}]$, then $Y = g(\boldsymbol{X}) + E$ where $E \sim \mathcal{N}(0, \sigma)$. Such a simple assumption seems too good to be true, but in fact it gives a great fit to the data (§4.1). Even more surprisingly, if you retrain the same hyperparameters many times, the scores do become normally distributed with homogeneous variance as you enter the asymptotic regime. In §4.2, we run precisely this experiment with ResNet18 and find the conditional distribution shows a high degree of normality with essentially constant variance for hyperparameter configurations in the top 62%. Thus, additive noise offers a realistic model for this random variation.

Therefore, we assume $Y = g(\boldsymbol{X}) + E$. From our prior analysis (§2.2), the tail of $g(\boldsymbol{X})$ converges to a quadratic distribution. Assuming $\sigma$ is small, if $Y = g(\boldsymbol{X}) + E$ is in the tail then so is $g(\boldsymbol{X})$, thus:

$$Y \approx Q + E, \qquad Q \sim \mathcal{Q}_{\max}(\alpha, \beta, \gamma), \ E \sim \mathcal{N}(0, \sigma) \tag{11}$$

To model $Y$, we define a new family. *The noisy quadratic distribution*, $\mathcal{Q}(\alpha, \beta, \gamma, \sigma)$, is the sum of a quadratic and a normal random variable. Like the quadratic distribution, it comes in two variants: the *concave* ($\mathcal{Q}_{\max}$) and *convex* ($\mathcal{Q}_{\min}$) noisy quadratic distributions. Moreover, when $\sigma = 0$, we recover the (noiseless) quadratic distribution as a special case.[3]

Let us derive the noisy quadratic's CDF and PDF. *The partial expectation from $a$ to $b$ is defined as:*

$$\mathbb{E}_a^b[Z] := \mathbb{P}(a \leq Z \leq b)\, \mathbb{E}[Z|a \leq Z \leq b] = \int_a^b z f_Z(z) dz \tag{12}$$

Since $Y$ is the sum of two independent random variables, $Q$ and $E$, we can apply the convolution formula for its CDF: $F_Y(y) = \mathbb{E}[F_Q(y - E)]$. After some calculus (§E.2), this yields:

$$F_Y(y) = \Phi\left(\frac{y - \alpha}{\sigma}\right) - \mathbb{E}_0^1\left[V^{\gamma/2}\right], \qquad V \sim \mathcal{N}\left(\frac{\beta - y}{\beta - \alpha}, \frac{\sigma}{\beta - \alpha}\right) \tag{13}$$

Similarly, we have the convolution formula for the PDF: $f_Y(y) = \mathbb{E}[f_Q(y - E)]$, which becomes:

$$f_Y(y) = \frac{\gamma}{2(\beta - \alpha)} \mathbb{E}_0^1\left[V^{\frac{\gamma-2}{2}}\right], \qquad V \sim \mathcal{N}\left(\frac{\beta - y}{\beta - \alpha}, \frac{\sigma}{\beta - \alpha}\right) \tag{14}$$

Thus, we can express the CDF and PDF of the noisy quadratic distribution in terms of properties of the normal distribution. Alternative forms for Equations 13 and 14 are possible; however, we have found the ones above most theoretically and computationally useful.

When $\gamma$ is even, $\mathbb{E}_0^1[V^{\gamma/2}]$ is a partial moment of the normal distribution. Its partial moments are well-studied, with a recursive formula available (Winkler et al., 1972); however, when $\gamma$ is odd, it is a partial *fractional* moment. While formulas exist for both partial and fractional moments of the normal (Winkler et al., 1972; Winkelbauer, 2014), to the best of the authors' knowledge no formula is known for partial fractional moments. To compute them, more advanced numerical methods are necessary. Adapting these numerical methods required considerable effort; as a result, it is impractical to describe them here due to space considerations. Still, full details are documented and available in our implementation[4]. In addition, we plan to describe them in a future publication.

In summary, we extend the quadratic distribution to a more general parametric family: *the noisy quadratic distribution*. When tuning the hyperparameters of a deep learning model, *maximizing* should cause the score distribution's *right* tail to approximately match a *concave* noisy quadratic, while *minimizing* should cause its *left* tail to match a *convex* noisy quadratic. See §B for formulas.

---

[3]When the variant is clear from context, we write the distribution unadorned: $\mathcal{Q}(\alpha, \beta, \gamma, \sigma)$. Similarly, we differentiate the quadratic, $\mathcal{Q}(\alpha, \beta, \gamma)$, and noisy quadratic, $\mathcal{Q}(\alpha, \beta, \gamma, \sigma)$, by the presence of $\sigma$.

[4](URL redacted)

## 3 EXPERIMENTAL SETUP

To test our theory, we need large samples of random search with different models. We obtain 1,024 iterations of random search with three models: DeBERTa (He et al., 2021), DeBERTaV3 (He et al., 2023), and ResNet18 (He et al., 2016). DeBERTa and DeBERTaV3 are pretrained transformers for traditional NLP tasks. DeBERTa pretrains with a generative objective, while DeBERTaV3 utilizes a discriminative one. We use data from Lourie et al. (2024) who fine-tuned these models on MultiNLI (Williams et al., 2018), a natural language inference benchmark. Lourie et al. (2024) used the same search space for both models, running 1,024 iterations of random search for each and tuning the learning rate, proportion of the first epoch for warmup, batch size, number of epochs, and dropout. The other model, ResNet18, is a classic convolutional architecture for computer vision (He et al., 2016). We pretrain ResNet18 on ImageNet (Russakovsky et al., 2015), an image classification benchmark. Using the FFCV library,[5] we trained ResNet18 with momentum SGD and a 1-cycle learning rate schedule. We ran 1,024 iterations of random search, tuning the learning rate, peak epoch, momentum, batch size, epochs, weight decay, label smoothing, and use of blurpool.

The search distributions for these models can be found in §C. Now, we will describe the details of each analysis.

**Assessing Goodness of Fit.** To assess goodness of fit, we compare the score distribution to the fitted noisy quadratic distribution. We estimate the score distribution using the empirical CDF (eCDF) and highest density LD confidence bands as recommended in Lourie et al. (2024), and we fit the noisy quadratic distribution to the tail via censored maximum spacing estimation (Cheng & Amin, 1983; Ranneby, 1984). The best threshold for the asymptotic regime was selected using visual diagnostics.

**Estimating and Extrapolating the Tuning Curve.** To explore our theory's practical application, we subsample 48 iterations of random search without replacement from the full 1,024 for each model. We use all 1,024 iterations to estimate the ground truth eCDF. For nonparametric estimates, we construct the eCDF and LD highest density confidence bands from the subsample, as in Lourie et al. (2024). For parametric estimates, we select the asymptotic regime via visual diagnostics using only the subsample, fit the noisy quadratic distribution to the tail via censored maximum spacing estimation (Cheng & Amin, 1983; Ranneby, 1984), and compute parametric confidence bands from the nonparametric ones as consonance regions (Easterling, 1976). We compute these via brute-force search with a grid of 64 log-spaced values for $\sigma$, 128 and 256 linearly spaced values for $\alpha$ and $\beta$.

**Testing Additive Normal Errors.** To test our assumptions about the errors, we took the ResNet18 results and picked the hyperparameters at the 12.5th, 25th, up to 100th percentile of performance. We retrained each configuration 128 times, letting the initialization, data order, and so on vary. In this way, we obtained large samples characterizing the score distributions for fixed hyperparameters.

## 4 TESTING THE THEORY

A theory is useful only in so far as it describes the world. Thus, we now ask this question.

### 4.1 ASSESSING GOODNESS OF FIT

Does our theory accurately describe random search? Our primary claim is the score distribution's tail converges to the noisy quadratic. Let us test that claim by seeing how they compare.

Figure 3 makes that comparison, plotting the ground truth against its theoretical form. Across three different scenarios, Figure 3 shows an excellent fit between the theoretical form and the empirical distribution. In each scenario, both the noisy quadratic's CDF and its median tuning curve closely adhere to the ground truth. At all times, they remain within the 95% confidence bands. Moreover, as theory predicts, the point estimates fit the ground truth almost perfectly in the asymptotic regime.

These results suggest our assumptions are satisfied, but more importantly they show the theory is actually useful. A big question in any asymptotic analysis is: just how *asymptotic* is it? Do practical scenarios approach the limit enough for asymptotics to matter? Figure 3 answers with a resounding yes. In each scenario, a sizable portion of the score distribution falls within the asymptotic regime:

---

[5] https://github.com/libffcv/ffcv (commit: 92eba2e)

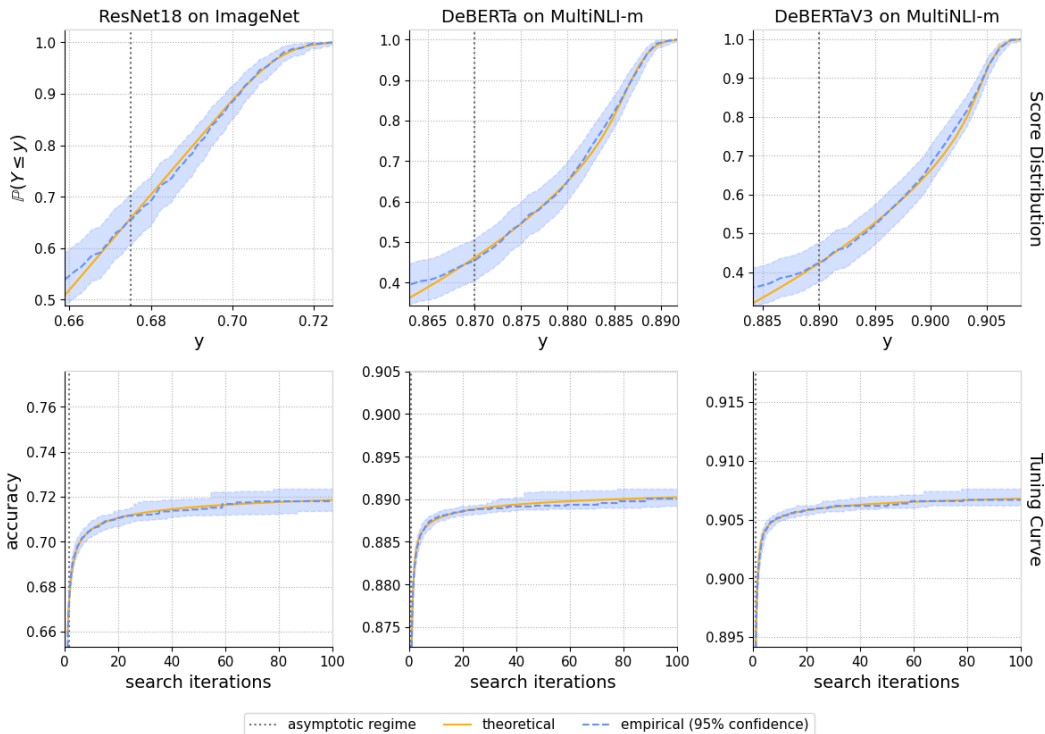

Figure 3: A comparison of the noisy quadratic (*theoretical*) and the score distribution (*empirical*). The top row depicts the CDFs, while the bottom depicts the tuning curves. Each column corresponds to a different scenario: pretraining ResNet18 on ImageNet, fine-tuning DeBERTa on MutliNLI, and fine-tuning DeBERTaV3 on MultiNLI. The *asymptotic regime* is the performance threshold above which the theoretical approximations apply. All estimates use the full 1,024 iterations of random search from each scenario. Empirical estimates are from the empirical distribution, while theoretical estimates use the noisy quadratic fitted to the tail via censored maximum spacing estimation.

34% for ResNet18, 54% for DeBERTa, and 57% for DeBERTaV3. The search spaces are designed to be large, characteristic of what practitioners might use when tuning these models on new problems. Still we see almost the entire tuning curve lies in the asymptotic regime. Thus, the asymptotic regime can be relevant in practical scenarios even from the first few iterations of random search.

A final note: in some sense, the models get more advanced as we move from ResNet18, to DeBERTa, to DeBERTaV3. ResNet18 is a convnet trained from scratch, DeBERTa is a pretrained transformer, and DeBERTaV3 makes further improvements. As the models become more advanced, more of the score distribution falls in the asymptotic regime. Perhaps this is why the asymptotic approximation is so effective? Models are engineered not only to be effective, but also easy to tune. Should this explanation be true, then our theory will only become more relevant as models continue to advance.

## 4.2 TESTING ADDITIVE NORMAL ERRORS

So, our theory describes random search, but does it really reflect what is actually happening? For example, if you fix the hyperparameters, are the scores normally distributed with constant variance?

To test this assumption we do just that. Our theory claims the scores become normal with constant variance as hyperparameters approach the asymptotic regime. The asymptotic regime is defined by a threshold on the score. Thus, we retrain hyperparameters with increasing scores. Namely, we take the random search results from ResNet18, pick the configurations at the 12.5th, 25th, 37.5th, up to the 100th accuracy percentiles, then retrain each 128 times, letting things like the initialization and data order vary. Thereby, we obtain large samples characterizing these distributions over the scores.

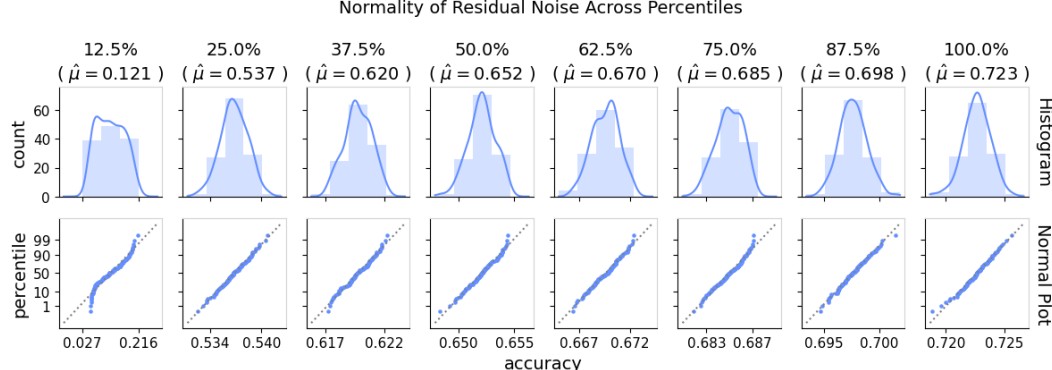

Figure 4: Diagnostic plots for the normality of the score distribution given fixed hyperparameters. The top row shows histograms with kernel density estimates, while the bottom shows Q-Q plots. Each column corresponds to the configuration at that accuracy percentile for ResNet18 on ImageNet. *All except the worst performing hyperparameters demonstrate normality to a very high degree.*

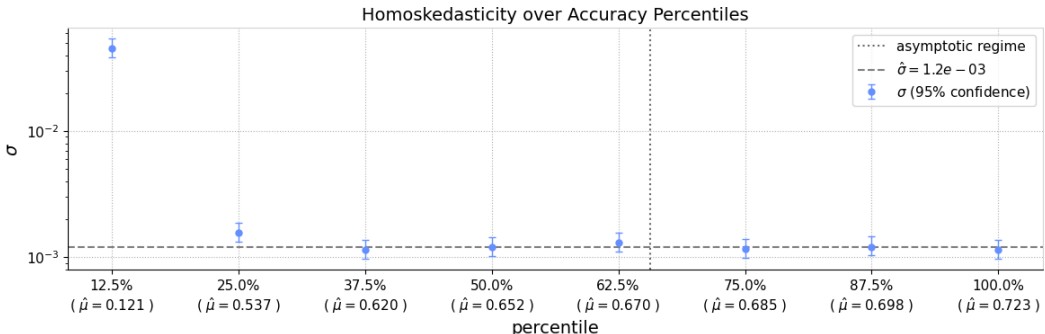

Figure 5: A comparison of standard deviations for score distributions given fixed hyperparameters. The $x$-axis displays configurations at different accuracy percentiles for ResNet18 on ImageNet. Each point estimates the standard deviation at that percentile. Confidence intervals are simultaneous, constructed using the $\chi^2$ interval for the standard deviation of the normal and a Šidák correction. *The standard deviations quickly converge to a constant long before the asymptotic regime.*

To test the additive normal errors assumption, we break it down into two parts: first, the scores are normally distributed; and second, their standard deviations are constant (i.e., *homoskedastic*).

**Testing Normality.** We test normality using the traditional normal probability plot. In addition, we show histograms and kernel density estimates (KDEs) to offer a more intuitive visualization. Figure 4 displays the results. On the top, the histograms and KDEs reveal that, except for the worst hyperparameters, the distributions exhibit that familiar bell curve. The normal probability plots are even more informative. On the bottom, the sample quantiles almost all fall on that $y = x$ line, equating them with the quantiles of the normal distribution and establishing their fit. Thus, each distribution from the 25th percentile and up achieves a high degree of normality.

**Testing Homoskedasticity.** We test homoskedasticity by plotting simultaneous confidence intervals for the standard deviations at the different accuracy percentiles. Since we have confirmed normality, we use the classic confidence interval for the standard deviation of a normal distribution; since the intervals are independent, we make them hold simultaneously using a Šidák correction. These simultaneous intervals then bound how different the standard deviations can be. Figure 5 shows the result. The standard deviation drops to a constant around the 37.5th percentile. From then on, all 95% confidence intervals contain a common value for it (e.g., 1.2e-3, the average of the last four). Moreover, as the intervals are so tight, it is unlikely any large differences exist. Thus, we see that the standard deviation starts off inflated, but converges to a constant long before the asymptotic regime.

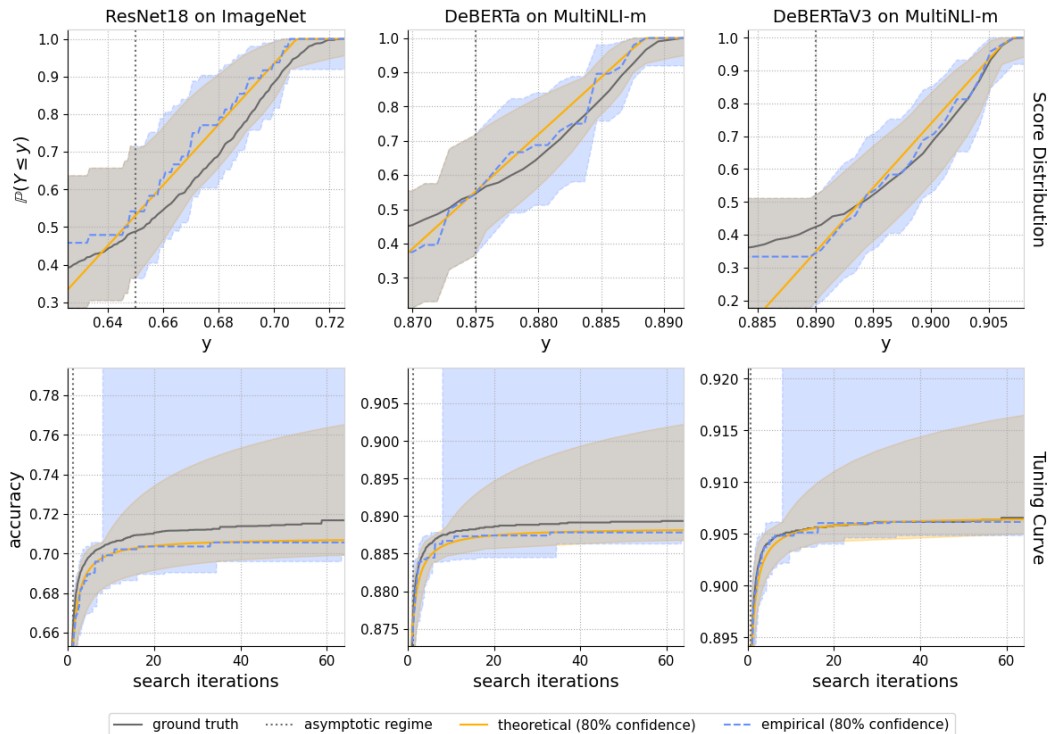

Figure 6: A comparison of noisy quadratic (*theoretical*) and nonparametric (*empirical*) estimates. The top row depicts the CDFs, while the bottom depicts the tuning curves. Each column corresponds to a different scenario: pretraining ResNet18 on ImageNet, fine-tuning DeBERTa on MultiNLI, and fine-tuning DeBERTaV3 on MutliNLI. The *asymptotic regime* is the performance threshold above which the noisy quadratic was fit. All estimates use 48 iterations of random search subsampled from that scenario's full 1,024. Empirical estimates are from the empirical distribution, while theoretical estimates use the noisy quadratic fitted to the tail via censored maximum spacing estimation.

Both experiments point to the same conclusion: bad hyperparameters exhibit bad structure, but as the hyperparameters improve—as you approach the asymptotic regime—simple structure emerges.

### 4.3 ESTIMATING AND EXTRAPOLATING THE TUNING CURVE

Beyond explaining random search, can we predict how it will progress? Let us look at estimating the noisy quadratic from a small number of samples, then using it to infer the tuning curve's shape.

To explore such a practical use case, we subsampled 48 iterations of random search in each of our three scenarios. Using each subsample, we plotted the eCDF to visually determine a threshold for the asymptotic regime. We chose thresholds based on where the eCDF begins to show a smoother structure. More generally, one could try several and choose based on the noisy quadratic's fit. In this way, we estimated the thresholds as 0.65 for ResNet18, 0.875 for DeBERTa, and 0.89 for DeBERTaV3. Then, we fit the noisy quadratic distribution to the subsample using the threshold.

Figure 6 compares the parametric estimates against their nonparametric baselines. The parametric point estimates mostly smooth out the nonparametric ones. Intuitively, this makes sense as both attempt to fit the same data without any kind of prior. Both estimate the tuning curve to varying degrees of precision across the scenarios. For ResNet18, the gap is about 1 point in accuracy, while for DeBERTa it is 0.2 points, and for DeBERTaV3 the curves are almost identical. This variable precision emphasizes the need for confidence bands—where the two approaches give very different results. Indeed, the parametric confidence bands dramatically tighten their nonparametric counterparts. While the nonparametric bands become trivial after 8 iterations, the parametric bands extrapolate beyond the entire 48 used in their construction, all while still enclosing the ground truth.

## 5 RELATED WORK

Hyperparameters have always been an essential part of deep learning. Today, researchers seek new ways to set them—both theoretical frameworks like $\mu$Transfer (Yang et al., 2021) and empirical ones such as scaling laws (Hestness et al., 2017; Rosenfeld et al., 2020; Kaplan et al., 2020; Henighan et al., 2020; Hernandez et al., 2021; Hoffmann et al., 2022). Much of the motivation comes from new challenges posed by foundation models; however, long before the current era, hyperparameters were still the subject of much practical advice (Orr & Müller, 1998; Montavon et al., 2012).

Often, this advice was highly effective but ad hoc—focusing on specific hyperparameters and ways to determine them (Mishkin & Matas, 2016; Smith, 2018). Such advice is difficult to adapt to new hyperparameters and changing contexts. Thus, researchers have sought a more systematic approach.

The simplest approach remains most popular: cross-validation (Stone, 1974). Automating it leads to *hyperparameter search*, with many such algorithms developed (Bergstra et al., 2011; Bischl et al., 2023). Early on, Bergstra & Bengio (2012) discovered random search is surprisingly effective, far outperforming grid search. Since then, many have explored more advanced approaches (Snoek et al., 2012; Swersky et al., 2014; Wistuba et al., 2015; Pedregosa, 2016; Olson et al., 2016; Falkner et al., 2018; Awad et al., 2021; Wistuba et al., 2022; Kadra et al., 2023); still, random search remains a strong baseline, with variants obtaining high performance (Li et al., 2018; 2020).

Hyperparameter search excels at finding a good model for production, but in research we face a different problem. Instead of constructing the best model, we want to understand a new idea or get an insight into some phenomenon—we want to answer questions, such as: are the hyperparameters tuned enough? Why is this model difficult to tune? And, what is the best possible performance?

We need tools tailored for research. Accepting the task, Dodge et al. (2019) proposed *the tuning curve*,[6] deriving the first point estimator for it. Soon after, Tang et al. (2020) identified the need for effective confidence bands, discovering that the default approach (the bootstrap) fails to achieve meaningful coverage. More recently, Lourie et al. (2024) developed such confidence bands. Their bands are simultaneous, exact, and distribution-free—making them quite robust; however, since the bands are nonparametric, they only bound the initial segment of the curve. They do not extrapolate. Enabling such extrapolation was the beginning motivation behind our theory of random search.

Our theory derives a limit for the score distribution's tail. Similar limits are often explored under *extreme value theory* (Coles, 2001; de Haan & Ferreira, 2006). For example, the Pickands-Balkema-De Haan theorem gives conditions under which the tail converges to a generalized Pareto distribution (Pickands, 1975; Balkema & de Haan, 1974). This distribution relates closely to the (noiseless) quadratic, though the noisy quadratic is distinct. In general, while extreme value theory seeks broad theorems with an abstract approach, we seek specific analyses based on beliefs about the underlying mechanism. Our aim is to build an empirical theory to better understand our deep learning models.

## 6 CONCLUSION

We derived an asymptotic theory of random search. The theory emerges from two ideas: the score is a smooth function of the hyperparameters, and its noise is normal with constant variance. Surprisingly, these assumptions describe deep learning quite accurately in practice.

Using the theory, we derived a parametric form for the score distribution's tail. As iterations increase, this tail determines random search's behavior. Thus, the parametric distribution can offer better point estimates and extrapolate confidence bands for tuning curves. The limiting distribution has two forms: the quadratic in the deterministic case, and the noisy quadratic in the stochastic one. The noisy quadratic generalizes the first, and has four interpretable parameters: $\alpha$, a measure of the probability in the asymptotic regime, $\beta$, the average performance of the best possible hyperparameters, $\gamma$, the effective number of hyperparameters, and $\sigma$, the standard deviation of the scores when you retrain with fixed hyperparameters.

Our theoretical framework offers a new set of tools for deep learning research. We hope they help practitioners build better models and researchers discover novel insights. Thus, we release a library making them available: (URL redacted).

---

[6]They introduced the concept, though the term *tuning curve* came later in Lourie et al. (2024).

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

## A  THE QUADRATIC DISTRIBUTION

As derived in §2.2, the formulas for the concave quadratic distribution are:

$$F(y; \alpha, \beta, \gamma) := 1 - \left( \frac{\beta - y}{\beta - \alpha} \right)^{\gamma/2} \tag{15}$$

$$f(y; \alpha, \beta, \gamma) = \frac{\gamma}{2(\beta - \alpha)} \left( \frac{\beta - y}{\beta - \alpha} \right)^{\frac{\gamma - 2}{2}} \tag{16}$$

The equivalent formulas for the convex quadratic distribution are:

$$F(y; \alpha, \beta, \gamma) := \left( \frac{y - \alpha}{\beta - \alpha} \right)^{\gamma/2} \tag{17}$$

$$f(y; \alpha, \beta, \gamma) = \frac{\gamma}{2(\beta - \alpha)} \left( \frac{y - \alpha}{\beta - \alpha} \right)^{\frac{\gamma - 2}{2}} \tag{18}$$

The quadratic distribution is supported only on the interval $\alpha \leq y \leq \beta$. These formulas are valid within that interval. Outside of it, the density is 0; below it, the CDF is 0; above it, the CDF is 1.

## B  THE NOISY QUADRATIC DISTRIBUTION

As derived in §2.3, the formulas for the concave noisy quadratic distribution are:

$$F_Y(y) = \Phi\left( \frac{y - \alpha}{\sigma} \right) - \mathbb{E}_0^1 \left[ V^{\gamma/2} \right], \qquad V \sim \mathcal{N}\left( \frac{\beta - y}{\beta - \alpha}, \frac{\sigma}{\beta - \alpha} \right) \tag{19}$$

$$f_Y(y) = \frac{\gamma}{2(\beta - \alpha)} \mathbb{E}_0^1 \left[ V^{\frac{\gamma - 2}{2}} \right], \qquad V \sim \mathcal{N}\left( \frac{\beta - y}{\beta - \alpha}, \frac{\sigma}{\beta - \alpha} \right) \tag{20}$$

The equivalent formulas for the convex noisy quadratic distribution are:

$$F_Y(y) = \Phi\left( \frac{y - \beta}{\sigma} \right) + \mathbb{E}_0^1 \left[ V^{\gamma/2} \right], \qquad V \sim \mathcal{N}\left( \frac{y - \alpha}{\beta - \alpha}, \frac{\sigma}{\beta - \alpha} \right) \tag{21}$$

$$f_Y(y) = \frac{\gamma}{2(\beta - \alpha)} \mathbb{E}_0^1 \left[ V^{\frac{\gamma - 2}{2}} \right], \qquad V \sim \mathcal{N}\left( \frac{y - \alpha}{\beta - \alpha}, \frac{\sigma}{\beta - \alpha} \right) \tag{22}$$

Unlike the quadratic distribution, the noisy quadratic is supported on the entire real line.

## C  MODELS AND SEARCH DISTRIBUTIONS

As described in §3, we use random search results for DeBERTa, DeBERTaV3, and ResNet18.

For DeBERTa and DeBERTaV3, Lourie et al. (2024) used the following search distribution:

$$\begin{aligned}
\texttt{batch\_size} &\sim \mathrm{DiscreteUniform}(16, 64) \\
\texttt{num\_epochs} &\sim \mathrm{DiscreteUniform}(1, 4) \\
\texttt{warmup\_proportion} &\sim \mathrm{Uniform}(0, 0.6) \\
\texttt{learning\_rate} &\sim \mathrm{LogUniform}(1e{-}6, 1e{-}3) \\
\texttt{dropout} &\sim \mathrm{Uniform}(0, 0.3)
\end{aligned}$$

Note that `warmup_proportion` is the proportion of the first epoch only.

For ResNet18, we used the following search distribution:

$$\text{epochs} \sim \text{DiscreteUniform}(20, 100)$$
$$\text{batch\_size} \sim \text{DiscreteUniform}(\{128, 256, 512, 1024\})$$
$$\text{lr} \sim \text{LogUniform}(5e{-}3, 5e1)$$
$$\text{lr\_peak\_epoch} = \lfloor \text{proportion} \times \text{epochs} \rfloor, \ \text{proportion} \sim \text{Uniform}(0, 0.8)$$
$$\text{momentum} \sim \text{Uniform}(0.7, 1.0)$$
$$\text{weight\_decay} \sim \text{LogUniform}(1e{-}6, 1e{-}3)$$
$$\text{label\_smoothing} \sim \text{Uniform}(0.0, 0.5)$$
$$\text{use\_blurpool} \sim \text{DiscreteUniform}(0, 1)$$

In Appendix D, we present additional results that validate our theory's generality and how it applies across architectures. For it, we run random search on AlexNet (Krizhevsky et al., 2012; Krizhevsky, 2014), ResNet18 (He et al., 2016), and ConvNext Tiny (Liu et al., 2022) using the search distribution above, except fixing `use_blurpool` to 0 because ConvNext does not use maxpool (or blurpool) layers and thus we can not consistently apply the hyperparameter to all three.

## D  GENERALIZATION ACROSS ARCHITECTURES

While our theory is general, it is also asymptotic; thus, it is natural to ask: how quickly does the asymptotic approximation apply in practice? For ResNet18, DeBERTa, and DeBERTaV3, we saw the asymptotic regime covered 34%, 54%, and 57% of the score distribution—applying from the first or second iteration of random search. Still, perhaps the asymptotic regime applies only because these architectures are so advanced, or the search spaces match them particularly well.

To investigate such questions, we compare ResNet18 with two other architectures: AlexNet (Krizhevsky et al., 2012; Krizhevsky, 2014) and ConvNext Tiny (Liu et al., 2022). AlexNet goes from ResNet into the past: many consider it the first major architecture of the current deep learning renaissance and, as such, it is considerably less advanced than ResNet—missing later innovations such as batch normalization or residual connections. On the other hand, ConvNext goes from ResNet into the future: it starts with the ResNet architecture and applies lessons learned from transformer-based models. We obtain 170, 495, and 162 iterations of random search for AlexNet, ResNet18, and ConvNext, using the same search distribution as before except fixing `use_blurpool` to 0 because blurpool is not compatible with ConvNext. Moreover, we use the same search distribution across all three models to guarantee it is not unusually well-suited to any specific one.

Figure 7 presents the results.

**The asymptotic regime is large in practice.**   Across all architectures, the asymptotic regime is more than large enough to be practically relevant. Of the search distributions, it covers 17% for AlexNet, 31% for ResNet18, and 43% for ConvNext Tiny. In other words, it characterizes the tuning curve after 2-4 iterations of random search. Thus, our theory describes random search with a realistic budget. The search distribution can not be the driving factor behind this result because we use the same one across all architectures. Moreover, while better architectures display larger asymptotic regimes (e.g., ResNet18 and ConvNext), our theory even describes an older less advanced architecture like AlexNet after just a handful of search iterations.

**The effective number of hyperparameters is stable across architectures.**   An interesting thing happens when we use the same search space across the different architectures: the effective number of hyperparameters ($\gamma$) remains constant. For AlexNet, ResNet, and ConvNext Tiny, the estimate of $\gamma$ is 2. DeBERTa and DeBERTaV3 exhibit a similar phenomenon: Lourie et al. (2024) used the same search space for both, and both models show $\gamma = 1$. These results suggest an intuitive conclusion: the effective number of hyperparameters seems to be more a property of the search space, i.e. the hyperparameters themselves. Thus, it exhibits some stability across models.

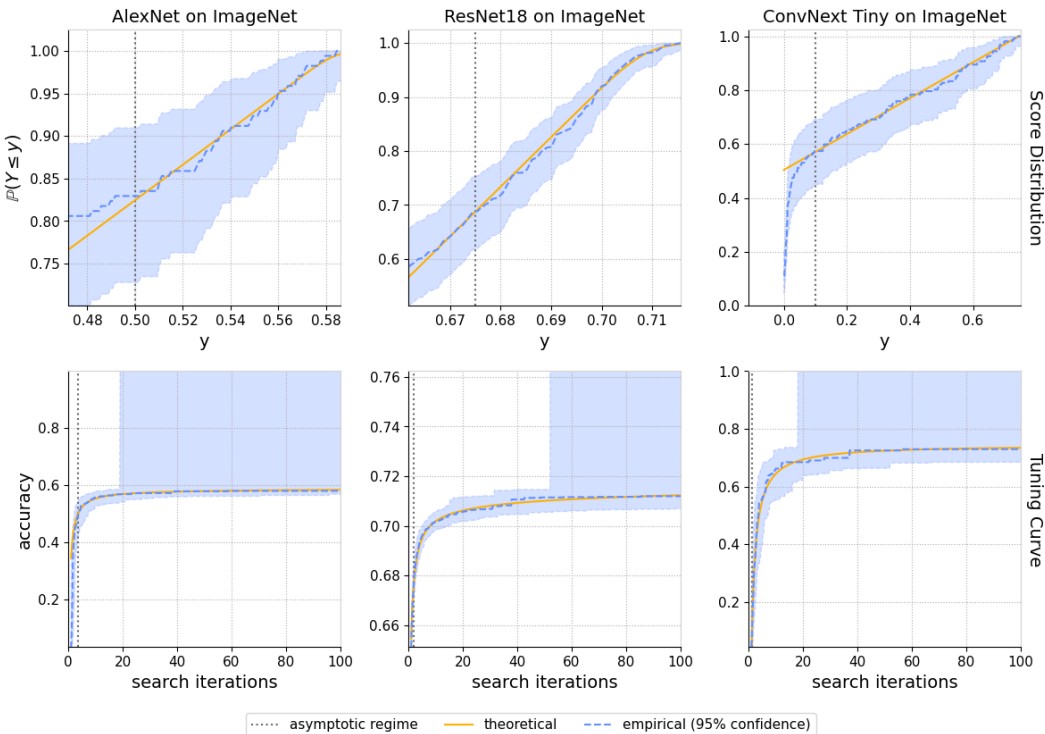

Figure 7: A comparison of the noisy quadratic (*theoretical*) and the score distribution (*empirical*) across different architectures trained on ImageNet. Each column corresponds to a different model: AlexNet, ResNet18, and ConvNext. All models use the same search distribution, and the estimates use 170, 495, and 162 iterations of random search, respectively. Empirical estimates are from the empirical distribution, while theoretical estimates use the noisy quadratic fitted to the tail via censored maximum spacing estimation.

**Convergence is not necessary.** In modern deep learning, training is often limited by compute. As a result, our theory must apply even when the network is not trained to convergence. Fortunately, the ConvNext Tiny results demonstrate this to be true. Despite its name, ConvNext Tiny is significantly larger than ResNet18 (29M vs 12M parameters)—instead, it is more comparable to ResNet50. As our training recipe was chosen for ResNet18, it does not use enough compute (epochs) for ConvNext Tiny to fully converge. This fact is evident in the best accuracy achieved: 73.6% as opposed to 82.1% in Liu et al. (2022). Even in this compute-limited regime, the theory still obtains an excellent fit.

**Better models have bigger asymptotic regimes.** Given the complexity of modern neural networks, one might ask: why can a simple theory describe their hyperparameters so well? One possible answer: because we designed them to be easy to optimize. Hyperparameter robustness is both a goal in itself and a side-effect of improving training. Should this be true, we would expect the asymptotic regime to grow over time—and that is exactly what we see. AlexNet has the smallest asymptotic regime, 17%; ResNet18 grows it to 31%, and ConvNext Tiny grows it to 43%. We saw a similar trend with DeBERTa (54%) and DeBERTaV3 (57%). Astoundingly, ConvNext actually grows the asymptotic regime in two ways: both as a percentage of the distribution, and as a range of accuracies. In the top right of Figure 7, we see the asymptotic regime span from 10% to 73.6% accuracy—a massive range of clear practical relevance. Since more advanced models have larger asymptotic regimes, this suggests the theory will only become more applicable over time.

## E PROOFS & THEOREMS

In §2, we derived our limits without emphasizing formality. We did this for two reasons. First, there are many ways to formalize the theorem—without a particular goal, any specific choice is

arbitrary. Second, whether the limit applies in practice is ultimately an empirical question. Consider the normal distribution: numerous versions of the central limit theorem exist, each applying in its own context. What is important is not that one set of conditions produces the normal distribution, but that many do. Therefore, we expect it might appear and, accordingly, use diagnostics like normal probability plots to determine if it has. That in mind, we now illustrate one way to formalize things.

### E.1 THE DETERMINISTIC CASE

We prove a limit theorem for minimization via random search in the deterministic case.

First, we need the following proposition, which gives a kind of inverse continuity near the minimum:

**Proposition E.1.** *Let $\mathbb{X} \subset \mathbb{R}^d$ be compact, $\mathbb{Y} \subset \mathbb{R}$, $g : \mathbb{X} \to \mathbb{Y}$ continuous, and $y_* = g(\boldsymbol{x}_*)$ its unique minimum. Then $\forall \delta > 0$, $\exists \epsilon$ such that $|g(\boldsymbol{x}) - y_*| < \epsilon$ implies $\|\boldsymbol{x} - \boldsymbol{x}_*\| < \delta$.*

*Proof.* For contradiction, assume $\delta > 0$ is such that the conclusion is false. Let $\epsilon_i$ be any sequence such that $\epsilon_i \to 0$. For each $\epsilon_i$, there exists some $\boldsymbol{x}_i$ such that $|g(\boldsymbol{x}_i) - y_*| < \epsilon_i$ but $\|\boldsymbol{x}_i - \boldsymbol{x}_*\| > \delta$, otherwise the conclusion would be true.

Consider the sequence $\boldsymbol{x}_i$. Since $\mathbb{X}$ is compact, it has a convergent subsequence: $\boldsymbol{x}_{i_k} \to \boldsymbol{x}_\infty$. By construction, $|g(\boldsymbol{x}_{i_k}) - y_*| < \epsilon_{i_k}$. As $\epsilon_i \to 0$, we have $g(\boldsymbol{x}_{i_k}) \to y_*$, and because $g$ is continuous:

$$g(\boldsymbol{x}_\infty) = g\left(\lim_{i_k \to \infty} \boldsymbol{x}_{i_k}\right) = \lim_{i_k \to \infty} g(\boldsymbol{x}_{i_k}) = y_*$$

However, $\|\boldsymbol{x}_{i_k} - \boldsymbol{x}_*\| > \delta$ so $\boldsymbol{x}_\infty \neq \boldsymbol{x}_*$, contradicting uniqueness of the minimum. $\square$

**Theorem E.2.** *Let $\mathbb{X} \subset \mathbb{R}^d$ be compact, $\mathbb{Y} \subset \mathbb{R}$, $g : \mathbb{X} \to \mathbb{Y}$ thrice continuously differentiable, $y_* = g(\boldsymbol{x}_*)$ its unique minimum in the interior of $\mathbb{X}$, $H_{\boldsymbol{x}_*}$ the Hessian at $\boldsymbol{x}_*$ having full rank, and $\boldsymbol{X} \sim \mathcal{X}$ a distribution over $\mathbb{X}$ with continuous PDF, $\mu(\boldsymbol{x})$. If $Y = g(\boldsymbol{X})$ is a random variable with CDF $F(y)$, there exists a quadratic distribution with CDF $Q(y)$ such that $\lim_{y \to y_*} F(y)/Q(y) = 1$.*

*Proof.* Write the 2nd order Taylor approximation of $g$ at $\boldsymbol{x}_*$ as $t(\boldsymbol{x}) = y_* + \frac{1}{2}(\boldsymbol{x} - \boldsymbol{x}_*)^T H_{\boldsymbol{x}_*}(\boldsymbol{x} - \boldsymbol{x}_*)$. Consider some neighborhood of $\|\boldsymbol{x} - \boldsymbol{x}_*\| < \delta$. By Proposition E.1, we can require $y$ be sufficiently close to $y_*$ to guarantee $\boldsymbol{x}$ is in it. Throughout the neighborhood, let $\epsilon$ be the Taylor approximation's worst case error:

$$t(\boldsymbol{x}) - \epsilon < g(\boldsymbol{x}) < t(\boldsymbol{x}) + \epsilon \tag{23}$$

Consider $F(y) = \mathbb{P}(Y \leq y)$. By Equation 23, $\mathbb{P}(t(\boldsymbol{x}) + \epsilon \leq y) \leq \mathbb{P}(g(\boldsymbol{x}) \leq y) \leq \mathbb{P}(t(\boldsymbol{x}) - \epsilon \leq y)$. We can write this equivalently as:

$$\mathbb{P}(t(\boldsymbol{x}) \leq y - \epsilon_1) \leq \mathbb{P}(g(\boldsymbol{x}) \leq y) \leq \mathbb{P}(t(\boldsymbol{x}) \leq y + \epsilon_1) \tag{24}$$

Let us analyze $\mathbb{P}(t(\boldsymbol{x}) \leq y)$.

We will need the fact that $\mathcal{X}$ is approximately uniform near $\boldsymbol{x}_*$. Let $c = \mu(\boldsymbol{x}_*)$. As $\mu$ is continuous, $\mu(\boldsymbol{x}) \to c$ as $\boldsymbol{x} \to \boldsymbol{x}_*$. Let $\eta$ be the maximum difference in the neighborhood:

$$c - \eta < \mu(\boldsymbol{x}) < c + \eta \tag{25}$$

In this sense, we can think of $\mathcal{X}$ as approximately uniform with density between $c \pm \eta$.

Returning to the Taylor approximation, $g$ is thrice continuously differentiable so the Hessian is real symmetric thus diagonalizable: $H_{\boldsymbol{x}_*} = U^T \Lambda U$, with $U$ an orthonormal matrix and $\Lambda = \mathrm{diag}(\lambda_1, \ldots, \lambda_d)$ the eigenvalues. Think of $U$ as a change of coordinates, $\boldsymbol{u} = U\boldsymbol{x}$. Since $U$ is orthonormal with $|\det U| = 1$, by the change of variables theorem the density of $\mathcal{X}$ in these new coordinates is still approximately $c \pm \eta$.

Finally, consider the event: $\mathbb{P}(t(\boldsymbol{x}) \leq y)$. In the coordinates $\boldsymbol{u}$, $H_{\boldsymbol{x}_*}$ is a diagonal matrix and $t(\boldsymbol{u}) = y_* + \frac{1}{2}\sum_{i=1}^d \lambda_i(u_i - u_{*i})^2$; therefore, $t(\boldsymbol{u}) \leq y$ defines an ellipse:

$$\sum_{i=1}^d \frac{\lambda_i}{2}(u_i - u_{*i})^2 \leq y - y_*$$

The volume of this ellipse is:

$$(y - y_*)^{d/2} \left( \frac{\pi^{d/2}}{\Gamma\left(\frac{d}{2} + 1\right)} \prod_{i=1}^{d} \sqrt{\frac{2}{\lambda_i}} \right)$$

Take all the terms that do not depend on $y$ as a constant, $C$. The volume is then: $C(y - y_*)^{d/2}$. The probability $\mathbb{P}(t(\boldsymbol{x}) \leq y)$ is the density integrated over this volume. The density is between $c - \eta$ and $c + \eta$, thus the probability is between products of these values and the volume:

$$C(y - y_*)^{d/2}(c - \eta) < \mathbb{P}(t(\boldsymbol{x}) \leq y) < C(y - y_*)^{d/2}(c + \eta) \tag{26}$$

Combining Equations 24 and 26, we have:

$$C(y - \epsilon - y_*)^{d/2}(c - \eta) < \mathbb{P}(g(\boldsymbol{x}) \leq y) < C(y + \epsilon - y_*)^{d/2}(c + \eta)$$

Using the parametrization of the (convex) quadratic distribution's CDF as $Q(y) = \omega(y - \alpha)^{\gamma/2}$, let $\omega = Cc$, $\alpha = y_*$, and $\gamma = d$. Then dividing by $Q(y)$ we have:

$$\frac{(c - \eta)}{c} \left( 1 - \frac{\epsilon}{y - y_*} \right)^{d/2} < \frac{F(y)}{Q(y)} < \frac{(c + \eta)}{c} \left( 1 + \frac{\epsilon}{y - y_*} \right)^{d/2} \tag{27}$$

Consider what happens as $y \to y_*$. By Proposition E.1 the neighborhood about $\boldsymbol{x}_*$ shrinks. As a result, $\eta \to 0$ and since $g$ is thrice differentiable the Taylor approximation's error goes to 0 at 3rd order while $y - y_*$ goes to 0 at 2nd order, thus $\epsilon/(y - y_*) \to 0$. Therefore, the upper and lower bounds in Equation 27 go to 1 and thus $F(y)/Q(y) \to 1$ as well. In other words:

$$\lim_{y \to y_*} \frac{F(y)}{Q(y)} = 1$$

$\square$

Thus, we obtain a limit theorem for random search under minimization, maximization being similar.

A few remarks are in order. We have shown convergence under one set of conditions; however, convergence can happen under other conditions as well. For example, we used uniqueness of the minimum to ensure that as $y$ approaches $y_*$, the corresponding $\boldsymbol{x}$ also approaches $\boldsymbol{x}_*$, the center of our Taylor approximation. If a finite number of distinct minima exist, this condition still holds as we approach the global minimum. Even with multiple global minima, they can be added together without issue. For example, the volume of their ellipses will be: $\sum_{j=1}^{n} C_j (y - y_*)^{d/2} = (y - y_*)^{d/2} \sum_{j=1}^{n} C_j$. As this example shows, many variants of the theorem are possible.

One assumption in particular merits deeper discussion: that the Hessian is full rank. Empirically, this assumption is rarely true. In all our experiments, the effective number of hyperparameters was fewer than the nominal number—in other words, the Hessian was rank deficient. Here is one way to close this gap: if $g$ is constant along the kernel of the Hessian, then you can marginalize over the kernel and consider $g$ as a function of the quotient space, in which the Hessian will have full rank.

In the end, we just need the hyperparameter loss to be approximately quadratic in some coordinates for which the search distribution is approximately uniform. Designing the search space so these assumptions are better satisfied will speed up convergence. For example, you can search for each hyperparameter using a uniform distribution on the appropriate scale (e.g., a log scale for the learning rate). Similarly, you can tighten the search space around the optimum so the Taylor approximation is a better fit.

### E.2 THE STOCHASTIC CASE

For the stochastic case, the noisy quadratic distribution is defined as the sum of a quadratic and a normal random variable. If the conditional mean $g(\boldsymbol{X}) = \mathbb{E}[Y|\boldsymbol{X}]$ satisfies the conditions of Theorem E.2, then it will converge to a quadratic distribution. If in addition $Y = g(\boldsymbol{X}) + E$, $E \sim \mathcal{N}(0, \sigma)$ then one just needs $\sigma$ to be small enough, otherwise the noise ($E$) will contaminate points where the quadratic distribution is a good approximation with the points where it is a bad one.

We just need to check the formulas for the CDF and PDF. We will show them for maximization.

**Proposition E.3.** *Let* $Y = Q + E$, *with* $Q \sim \mathcal{Q}_{\max}(\alpha, \beta, \gamma)$ *and* $E \sim \mathcal{N}(0, \sigma)$. *If* $F_Y(y)$ *is the CDF of* $Y$ *then:*

$$F_Y(y) = \Phi\left(\frac{y - \alpha}{\sigma}\right) - \mathbb{E}_0^1\left[V^{\gamma/2}\right], \qquad V \sim \mathcal{N}\left(\frac{\beta - y}{\beta - \alpha}, \frac{\sigma}{\beta - \alpha}\right)$$

*Proof.* Let $F_Q(y)$ denote the CDF of $Q$. By the convolution formula for the CDF of a sum we have: $F_Y(y) = \mathbb{E}[F_Q(y - E)]$. Note that this expectation is taken over the normal variable, $E$. Recall:

$$F_Q(y) = \begin{cases} 0 & y < \alpha \\ 1 - \left(\frac{\beta - y}{\beta - \alpha}\right)^{\gamma/2} & \alpha \le y \le \beta \\ 1 & y > \beta \end{cases}$$

Then, using properties of expectations, we have:

$$F_Y(y) = \mathbb{E}[F_Q(y - E)]$$

$$= \mathbb{E}_{-\infty}^{y-\beta}[1] + \mathbb{E}_{y-\beta}^{y-\alpha}\left[1 - \left(\frac{\beta - (y - E)}{\beta - \alpha}\right)^{\frac{\gamma}{2}}\right] + \mathbb{E}_{y-\alpha}^{\infty}[0]$$

$$= \mathbb{E}_{-\infty}^{y-\alpha}[1] - \mathbb{E}_{y-\beta}^{y-\alpha}\left[\left(\frac{\beta - (y - E)}{\beta - \alpha}\right)^{\frac{\gamma}{2}}\right]$$

$$= \Phi\left(\frac{y - \alpha}{\sigma}\right) - \mathbb{E}_{y-\beta}^{y-\alpha}\left[\left(\frac{E + (\beta - y)}{\beta - \alpha}\right)^{\frac{\gamma}{2}}\right]$$

where $\Phi$ is the standard normal distribution's CDF. Applying the change of variables defined by:

$$V = \frac{E + (\beta - y)}{\beta - \alpha} \tag{28}$$

We obtain the desired formula:

$$F_Y(y) = \Phi\left(\frac{y - \alpha}{\sigma}\right) - \mathbb{E}_0^1\left[V^{\gamma/2}\right] \tag{29}$$

$\square$

**Proposition E.4.** *Let* $Y = Q + E$, *with* $Q \sim \mathcal{Q}_{\max}(\alpha, \beta, \gamma)$ *and* $E \sim \mathcal{N}(0, \sigma)$. *If* $f(y)$ *is the PDF of* $Y$ *then:*

$$f_Y(y) = \frac{\gamma}{2(\beta - \alpha)}\mathbb{E}_0^1\left[V^{\frac{\gamma - 2}{2}}\right], \qquad V \sim \mathcal{N}\left(\frac{\beta - y}{\beta - \alpha}, \frac{\sigma}{\beta - \alpha}\right)$$

*Proof.* Let $f_Q(y)$ denote the PDF of $Q$. By the convolution formula for the PDF of a sum we have: $f_Y(y) = \mathbb{E}[f_Q(y - E)]$. Note that this expectation is taken over the normal variable, $E$. Recall:

$$f_Q(y) = \begin{cases} 0 & y < \alpha \\ \frac{\gamma}{2(\beta - \alpha)}\left(\frac{\beta - y}{\beta - \alpha}\right)^{\frac{\gamma - 2}{2}} & \alpha \le y \le \beta \\ 0 & y > \beta \end{cases}$$

Then, using properties of expectations, we have:

$$f_Y(y) = \mathbb{E}[f_Q(y - E)]$$

$$= \mathbb{E}^{y-\beta}_{-\infty}[0] + \mathbb{E}^{y-\alpha}_{y-\beta}\left[\frac{\gamma}{2(\beta - \alpha)}\left(\frac{\beta - (y - E)}{\beta - \alpha}\right)^{\frac{\gamma-2}{2}}\right] + \mathbb{E}^{\infty}_{y-\alpha}[0]$$

$$= \frac{\gamma}{2(\beta - \alpha)}\mathbb{E}^{y-\alpha}_{y-\beta}\left[\left(\frac{\beta - (y - E)}{\beta - \alpha}\right)^{\frac{\gamma-2}{2}}\right]$$

$$= \frac{\gamma}{2(\beta - \alpha)}\mathbb{E}^{y-\alpha}_{y-\beta}\left[\left(\frac{E + (\beta - y)}{\beta - \alpha}\right)^{\frac{\gamma-2}{2}}\right]$$

Applying the change of variables defined by:

$$V = \frac{E + (\beta - y)}{\beta - \alpha} \tag{30}$$

We obtain the desired formula:

$$f_Y(y) = \frac{\gamma}{2(\beta - \alpha)}\mathbb{E}^1_0\left[V^{\frac{\gamma-2}{2}}\right] \tag{31}$$

$\square$

