# OpenReview forum: "An Asymptotic Theory of Random Search for Hyperparameters in Deep Learning"
_ICLR.cc/2025/Conference — Submitted to ICLR 2025_

### Official Review · Reviewer_NG9w · 2024-11-01

**Soundness:** 2
**Presentation:** 3
**Contribution:** 2
**Rating:** 3
**Confidence:** 3

**Summary:**

In this paper, the probability function relationship between random search and model performance is analyzed theoretically, and the corresponding parameterized distribution is designed. The effect of parameter estimation of this distribution is better than that of nonparametric estimation. With this distribution, you can have a good understanding of the impact of parameter adjustment under the task. It helps researchers to evaluate the self-designed method and modify the corresponding strategy without eliminating the influence of parameter adjustment.

**Strengths:**

In this paper, a new parameter distribution is proposed, which theoretically fits the model performance changes under random search parameters. The fitted curves can help researchers to do the next step, such as judging whether their model can solve the task based on the best predictions.

The three parameters in the new parameter distribution correspond to the actual parameter meanings, and the influence of the parameters can be roughly understood directly through the estimated distribution.

**Weaknesses:**

The task model in the experiment is slightly thin and not comprehensive. For example, in line 120, the authors mentioned "Which architecture", but in experiments, no choice of ResNet has been considered. The authors use ResNet18 directly.

The new parameter distribution proposed by the author is compared only with the non-parametric distribution, not with other simple parameter distributions, whether the simple parameter model is sufficient to approximate the ground truth.

The author claims to propose an asymptotic theory of random search, but in fact the author relies only on analysis rather than proof to provide an approximation, without any theoretical guarantee like asymptotic convergence or probably approximately correct (PAC) learning theory.

**Questions:**

See above

---

> ### Author Response · Authors · 2024-11-27
>
> Thank you for observing the practical utility of our tool and how it can help researchers in deciding “the next step, such as judging whether their model can solve the task”. As you remark, our theory enables researchers to estimate meaningful parameters that tell them about their problem, such as the best possible performance. It is exactly this use case&mdash;better tools for deep learning research&mdash;that is the goal of our work.
>
> You also raise several concerns, which we have addressed in the updated version of the paper.
>
> First, you mention we could broaden our experiments, in particular we could explore architectural variations. In fact, we do explore an architectural variant of ResNet in our original experiments: whether to use blurpool or maxpool layers (see the search distribution described in Appendix C). Still, we could explore major architectural variations as well as minor ones. Thus, we have included new results that analyze AlexNet [1] and ConvNext [2] using the same hyperparameter search distribution and task as for ResNet. Appendix D presents these results. Notably, our theory describes the outcomes from each of these architectures well, with the asymptotic approximation matching the data after 2-4 iterations of random search.
>
> For comparing to simpler parametric distributions, we prefer the distribution derived from our limit theorem for two reasons. The limit theorem gives us confidence that the distributional form is correct and thus should extrapolate. More importantly and as you point out, our distribution’s parameters have meaningful interpretations in terms of the original problem; therefore, their estimates are more informative to the researcher.
>
> Finally, you mention that while we propose a theory, our derivation was informal. On your suggestion, we have updated the paper with formal theorems and proofs (Appendix E). We kept the derivation in the main text informal for ease of presentation; however, the proofs in the appendix provide all the technical details and specify exactly the sense in which the score distribution converges to the noisy quadratic.
>
> [1]: A. Krizhevsky, I. Sutskever, G. E. Hinton. (2012). ImageNet Classification with Deep Convolutional Neural Networks. Advances in Neural Information Processing Systems, 25.
>
> [2]: Z. Liu, H. Mao, C. -Y. Wu, C. Feichtenhofer, T. Darrell and S. Xie. (2022). A ConvNet for the 2020s. IEEE/CVF Conference on Computer Vision and Pattern Recognition (CVPR), pp. 11966-11976.

---

### Official Review · Reviewer_FayV · 2024-11-01

**Soundness:** 2
**Presentation:** 3
**Contribution:** 3
**Rating:** 5
**Confidence:** 3

**Summary:**

The paper describes a theory of tuning curves for random hyper-parameter tuning near the optimum under a smoothness assumption.
A parametric form of the tuning curve is described based on a novel description of the distribution of outcomes, and confirmed on three deep learning models.

**Strengths:**

Hyper-parameter tuning is still a critical aspect of deep learning research and practice, and is understudied in the ML community.
The paper proposes a concise methodology to understand the asymptotics of randomized search, with clear predictive capabilities.

The paper is well structured and clearly written.

**Weaknesses:**

I'm skeptical of a core assumption of the paper, which is whether the asymptotic regime is relevant in practice. If the function is  smooth, and well-approximated by a quadratic locally, then clearly random search is not the right tool. Using a GP would provide immense benefits if the local smoothness assumption holds, and the search is "near the optimum".
The main reason that random search is so successful is that in practice, many areas of the search space are not smooth, and jumps are common.
I'm quite surprised by how smooth the tuning curves in figure 1 and 3 are, and they are very unlike tuning curves I have seen with random search, which often stay constant for a long time, and don't progress for 10 or more iterations.

This might be due to the architectures used being extremely well understood, and, potentially, as the authors point out, easy to tune.
I would be quite curious to see if these results hold when tuning an MLP on, say the AutoML benchmark, or TabZilla.

Given the smoothness observed in the experiments given in this paper, I would be very interested to see how the tuning curve for TPE or a GP would look in these cases.

I did not study all the mathematics in detail. Given the assumptions the formulation seems reasonable; my main concern is about the assumptions and practical utility of the tool.

**Questions:**

How are the empirical confidence bands estimated in Figures 1 and 3?

---

> ### Comment · Reviewer_FayV · 2024-11-26
>
> After reading the other reviews, in particular reviewer TdXJ pointing out that random search never narrows the search-space, and therefore the asymptotic regime is unlikely to be the relevant one at any stage of the optimization, I'm adjusting my rating to "reject".

---

> > ### Author Response · Authors · 2024-11-27
> >
> > As discussed in our response to Reviewer TdXJ, their comment about lack of convergence is based on a misunderstanding of what was meant by “the region of relevant hyperparameters”. The sentence before the line they quote clarifies this: the region of relevant hyperparameters consists of “the ones better than the best you have seen so far”. Thus, it is not a function of the tuning algorithm, but rather how far tuning has progressed. This region converges to the optimum under general conditions: we have added a proof of this (Proposition E.1) and a formal proof of our limit theorem in Appendix E.
> >
> > Earlier, you also asked a related question: why are the tuning curves so smooth? As you point out, random search progresses in jumps, therefore a single run will be noisy with large flat regions. Instead of a single run, we estimate the *median* of all such runs over the search distribution. That is why the tuning curves in Figure 1 and 3 are smooth&mdash;the *probability* of finding a good configuration increases with each iteration.

---

> > > ### Comment · Reviewer_FayV · 2024-12-03
> > >
> > > Thank you for the clarification. Indeed, there was a misunderstanding on the distribution under investigation. P(Y>y) indeed converges. What is still surprising is how fast it converges in your experiments; this is highly counter-intuitive, as is the low dimensionality of the effective number of hyper-parameters.
> > > Thank you for including the additional experiments, indeed these make the empirical results much more compelling in my view.
> > >
> > > Can you explain how the median of all runs (i.e. ground truth in Figure 6) is computed from the results? Each ordering of the 1024 runs provides a different curve, correct? It's not immediately obvious how to compute the probability of improvement from this.
> > >
> > > Similarly, for 4.3, it's a bit unclear to me how the model was fit to the subset of 48 iterations, since again each ordering of the iterations would give a different tuning curve.
> > >
> > > Even 48 iterations would be a substantial time to wait for an initial experiment, and for such a budget, a more advanced tuning strategy is likely beneficial.
> > >
> > > I'm changing my rating back to my original rating given the clarification. I'm not yet convinced of the practicality of the approach, but I'm happy to adjust my rating based on the author response.

---

> > > > ### Author Response · Authors · 2024-12-04
> > > >
> > > > Thank you for reviewing our updates! We’re glad you found the additional evidence compelling. Your questions highlight some important points, and we will make sure to discuss them in our paper.
> > > >
> > > > > What is still surprising is how fast [the asymptotic approximation] converges in your experiments; this is highly counter-intuitive, as is the low dimensionality of the effective number of hyper-parameters.
> > > >
> > > > This counter-intuitiveness probably comes from how the use case changes the context. AutoML often considers complex search spaces with the goal of building the best model. In contrast, our work considers how best to analyze a common kind of deep learning experiment&mdash;the kind where a researcher compares a new model against a baseline. In this setting, the search space is more regular and the asymptotic approximation converges quickly; however, it’s possible (perhaps even expected) that convergence would be different in the AutoML setting. As this setting is not the focus of our investigation, it's left to future work. We'll make sure to mention this under the limitations.
> > > >
> > > > > Can you explain how the median of all runs (i.e. ground truth in Figure 6) is computed from the results? Each ordering of the 1024 runs provides a different curve, correct?
> > > >
> > > > Correct, each ordering produces a different curve; thus, we follow prior work and estimate the *pointwise* median, separately at each x-value. The resulting curve has the following interpretation: if you evaluate x hyperparameter configurations, then you have a 50% chance of doing better than y.
> > > >
> > > > To compute the ground truth, we use the method proposed in [1] for estimating median tuning curves. Intuitively, it:
> > > >
> > > > 1. Resamples the search iterations with replacement to produce many replicates.
> > > > 2. Selects the validation score in the i’th position from each replicate.
> > > > 3. Takes the median of these i’th positions across all the replicates.
> > > >
> > > > It turns out you can compute this without brute-force. A full description is provided in [1], but we’ll summarize it. If $Y_i$ is the score from the i’th iteration of random search, then we denote the best score after $k$ rounds by: $T_k = \max_{i=1}^k Y_i$. We have the following fact about $T_k$:
> > > >
> > > > $$\mathbb{P}(T_k\leq y) = \mathbb{P}\left(\max_{i=1}^k Y_i\leq y\right) = \mathbb{P}(Y_1\leq y\land\ldots\land Y_k\leq y)$$
> > > >
> > > > Since each round of random search is independent and identically distributed this equals:
> > > >
> > > > $$\mathbb{P}(T_k\leq y)=\prod_{i=1}^k\mathbb{P}(Y_i\leq y)=\mathbb{P}(Y\leq y)^k$$
> > > >
> > > > So, the median of $T_k$ is the value such that $\mathbb{P}(Y\leq y)^k = 0.5$. Letting $F(y) = \mathbb{P}(Y\leq y)$ and solving for $y$ gives: $y = F^{-1}(0.5^{1/k})$. We then let $F$ be the distribution from resampling with replacement (the *empirical distribution*).
> > > >
> > > > Note that the search iterations’ order doesn’t actually matter. Since random search samples hyperparameters independently, the results are just a sample from some fixed distribution.
> > > >
> > > > > Similarly, for 4.3, it's a bit unclear to me how the model was fit to the subset of 48 iterations, since again each ordering of the iterations would give a different tuning curve.
> > > >
> > > > Since the iterations’ order doesn’t matter in random search, we sampled 48 iterations without replacement from the full 1,024. The ground truth is still estimated with all 1,024 iterations, but the theoretical and empirical estimates only use the subsampled 48. The empirical estimate uses the 48 iterations to estimate the empirical distribution as in [1]. The theoretical estimate uses the 48 iterations to fit the noisy quadratic distribution. Both then compute the tuning curve via the formula $y = F^{-1}(0.5^{1/k})$ (see Section 3 of the paper for more details).
> > > >
> > > > > Even 48 iterations would be a substantial time to wait for an initial experiment, and for such a budget, a more advanced tuning strategy is likely beneficial.
> > > >
> > > > The main advantage of random search is that it runs in parallel. With enough compute, 48 iterations of random search takes the same time as 1. In this setup, random search actually takes *less* time than any sequential method. Of course, more compute isn’t free; however, we can dramatically reduce the compute cost for *experiments* using scaling laws. For example, the GPT-4 Technical Report [2] successfully extrapolated performance from models trained with 1/1,000th the compute. This is why our analysis is so well-suited for experiments: when tuning a large model for production, compute efficiency is very important; however, when exploring ideas in experiments, time efficiency and reproducibility are more important.
> > > >
> > > > [1] Lourie, N., Cho, K., & He, H. (2024). Show Your Work with Confidence: Confidence Bands for Tuning Curves. In Proceedings of the 2024 Conference of the North American Chapter of the Association for Computational Linguistics: Human Language Technologies. (pp. 3455–3472). ACL.
> > > >
> > > > [2] OpenAI. (2023). GPT-4 Technical Report. arXiv. https://arxiv.org/abs/2303.08774

---

> ### Author Response · Authors · 2024-11-27
>
> Thank you for highlighting the “clear predictive capabilities” of our theory and its “novel description of the distribution of outcomes” from random search. While you note our theory is “confirmed on three deep learning models”, you also raise two important concerns: are the theory’s assumptions satisfied more generally? And is it practically useful?
>
> For practical utility, the goal of our work is not to propose a state-of-the-art hyperparameter tuning algorithm. Clever and effective algorithms for this purpose already exist. Instead, we seek better tools for the design and analysis of deep learning experiments. Here, there are far fewer options. During the exploratory stage, researchers need to iterate quickly. As a result, they typically evaluate a single batch of hyperparameter configurations in parallel. Often, the hyperparameters are not fully tuned until a later stage. Our theory provides the statistical foundation necessary to analyze these results and determine if it is worth tuning more—perhaps even with a better tuning algorithm. Section 4.3 demonstrates this use of our theory by extrapolating the tuning curve.
>
> Besides new statistical methods, the theory also provides a better understanding of random search&mdash;and its limitations. As you mention, model-based optimization can have significant advantages. Our theory clarifies the contexts in which these advantages are most important. It has long been known that random search’s effectiveness relates to the number of *important* hyperparameters [1]. Our theory quantifies this relationship and formalizes it as *the effective number of hyperparameters*, $\gamma$. This parameter determines the shape of the noisy quadratic and how fast random search will progress. When $\gamma$ is too high, random search will be very slow. The surprising fact is that often $\gamma$ is low&mdash;in our experiments, it was always 1 or 2.
>
> Finally, you asked if our theory’s assumptions are satisfied more generally. We have added formal proofs in Appendix E, but ultimately this is an empirical question. In our experiments, we demonstrated our theory matches the outcomes from random search after the first 1-2 iterations. Those experiments examine 3 different models from both language and vision. Still, you brought up a valuable point: could the asymptotic theory fit because the models we consider are well understood? There are two ways in which this might happen: 1.) the architectures are particularly robust to their hyperparameters, or 2.) we know good search spaces for them.
>
> To address both concerns, we have added Appendix D: Generalization Across Architectures. In it, we use the *same* hyperparameter search distribution as for ResNet18, but apply it to AlexNet [2] and ConvNext [3]. AlexNet is an older architecture and thus much less developed than ResNet. In contrast, ConvNext is newer and more advanced. Together, these architectures span a decade of research. By using the same search space, we guarantee that it is not unusually well-suited to each. In this setting, our theory still matches the outcomes of random search from the first 2-4 iterations. As you may have expected, the least advanced architecture, AlexNet, needs more iterations for the asymptotic regime to become applicable; however, 4 iterations is still well within the bounds of practical relevance. More importantly, as newer architectures will be even more advanced, our theory only becomes *more* relevant over time.
>
> With these additions, our empirical results cover 5 architectures including convnets and transformers, from both vision and language, involving pretraining and finetuning, and spanning a decade of architectural improvements. Since we use the same search space across 3 models, it can not be tailored to each. In all these experiments, our theory describes random search after just a handful of iterations. We would be excited to see future work build off this foundation to analyze more advanced algorithms, like Bayesian optimization; still, our theory accurately describes a hyperparameter tuning method which is extremely common in practice.
>
> [1]: J. Bergstra, Y. Bengio. (2012).  Random Search for Hyper-Parameter Optimization. Journal of Machine Learning Research. 13(10):281−305.
>
> [2]: A. Krizhevsky, I. Sutskever, G. E. Hinton. (2012). ImageNet Classification with Deep Convolutional Neural Networks. Advances in Neural Information Processing Systems, 25.
>
> [3]: Z. Liu, H. Mao, C. -Y. Wu, C. Feichtenhofer, T. Darrell and S. Xie. (2022). A ConvNet for the 2020s. IEEE/CVF Conference on Computer Vision and Pattern Recognition (CVPR), pp. 11966-11976.

---

### Official Review · Reviewer_m19U · 2024-11-04

**Soundness:** 3
**Presentation:** 3
**Contribution:** 2
**Rating:** 5
**Confidence:** 3

**Summary:**

This paper proposes a simplified but accurate statistical model of hyperparameter tuning under random search in the "asymptotic" regime. Here the term "asymptotic" refers to hyperparameter settings which are "close" to optimal, for which a second-order taylor expansion around the set of optimal hyperparameters is informative. Fore this regime the authors heuristically propose the "quadratic" distribution to model the performance of random search in expectation over training randomness. To model the noisy effect from training randomness the authors propose a homoskedastic additive gaussian noise process which results in the "noisy quadratic" distribution. Over a variety of tasks the authors demonstrate the efficacy of this distribution for modeling random hyperparameter search.

**Strengths:**

The authors provide a clean and empirically compelling model of random hyperparameter search using a heuristic, first-principles based approach. The paper is written clearly, with a large amount of statistical and empirical validations.

**Weaknesses:**

It is quite unclear what the implications of these observations are. Also the framework only applies to random search as opposed to other randomized search methods such as Bayesian optimization. Currently the results seems like a few nice observations, but not a substantially impactful contribution.

**Questions:**

Is it possible to fit a model of H and perform a type of PCA procedure to determine the effective hyperparameters? Such an insight might help reduce the effective search space for certain classes of problems / hyperparameters. Are there any implications for other non-uniform random search methods such as Bayesian optimization? Or when doing muTransfer?

---

> ### Author Response · Authors · 2024-11-27
>
> Thank you for observing that our first-principles approach leads to a “clean and empirically compelling model of random hyperparameter search”, as well as our “large amount of statistical and empirical validations” that “demonstrate the efficacy of this distribution for modeling random hyperparameter search”. To provide even more evidence, we have complemented the paper’s informal derivation with formal proofs in Appendix E.
>
> You bring up a great point: while our theory provides a deeper understanding of random search, what is the practical impact? Random search is not the most efficient tuning algorithm; however, our goal is not to develop efficient tuning algorithms, but rather to provide better tools for the design and analysis of deep learning experiments.
>
> In experiments, researchers need to iterate quickly so they often evaluate a single batch of hyperparameter configurations in parallel. If an idea is obviously better or worse than the baseline, then hyperparameters need not be fully tuned at this exploratory stage. As a result, grid search and random search are quite common. Our theory enables these practitioners to estimate how much more performance might increase if they kept tuning—for example, if they decided to use a better tuning algorithm. In Section 4.3, we demonstrate this application by extrapolating tuning curves. Based on our proofs (Appendix E) and empirical results (Section 4 and Appendix D), our theory provides a statistically rigorous foundation for such analyses.
>
> In addition, though random search is not state-of-the-art, it remains a common hyperparameter tuning algorithm. Our asymptotic theory identifies the key determinants of random search’s performance, and predicts how it will progress based on them. The most important one is the effective number of hyperparameters: $\gamma$. When $\gamma > 3$, random search will not be very effective. In this way, our analysis provides a better understanding of random search and, in particular, its limitations. These limitations explain when and why advanced algorithms like Bayesian optimization outperform random search. While these algorithms are out of scope for our current work, it would be interesting for future work to extend the theory and analyze this more complicated case.

---

> ### Author Response · Authors · 2024-11-27
>
> You also brought up an interesting question: “Is it possible to fit a model of H and perform a type of PCA procedure to determine the effective hyperparameters?”. We had this same thought and explored this direction. One challenge is that you have to fit the Hessian, which has $O(d^2)$ parameters. For example, if there are 8 hyperparameters then the Hessian has 36 parameters. Combined with the noise due to random seeds and the need to only use points near the optimum, applying PCA to the Hessian can be difficult. Actually, this is one reason why our theory is useful: by taking a marginal approach, you can identify the asymptotic regime and the effective number of hyperparameters without having to fit the Hessian. Regardless of how many hyperparameters you consider, the noisy quadratic distribution has only 4 parameters to fit.

---

### Official Review · Reviewer_TdXJ · 2024-11-05

**Soundness:** 2
**Presentation:** 2
**Contribution:** 1
**Rating:** 3
**Confidence:** 3

**Summary:**

The authors propose to parametrize the performance of random search (tuning curve) with a noisy quadratic distribution. The authors test the fit and extrapolation of the proposed work in three experimental settings with diverse deep learning models.

**Strengths:**

-

**Weaknesses:**

- No baselines are considered.
- Limited number of experiments conducted. To really validate the claims of the paper one must consider diverse search spaces and models.
- The code for the work is not provided.
- The related work section is outdated.

**Questions:**

- **Line 180, "Thus as the search continues, the region of relevant hyperparameters converges about the optimum"**

   I do not agree with the above statement, random search, as the name gives, samples hyperparameters randomly. It is not a model-based method that incorporates the results into it's sampling stategy. So the region of relevant hyperparameters is the same search space (except maybe what was sampled before), it is not constrained in any manner. Additionally, being close to the optimum, would require a very very large number of trials in a continuous search space of $D$ dimensions.

- **The work defines the asymptotic regime as the hyperparameters that we care about the most, those close to the optimum (Line 81).**

   Looking at Figures 3 and 4, this perspective does not correspond to the explanation provided by the authors. For example, at the bottom of Figure 4 (the ResNet model), the region pointed out as the asymptotic regime in my perspective, would be somewhere at iteration 8-10. Random search there seems to be close to finding an optimum solution. While the asymptotic regime pointed out by the authors is around iteration 1-2.

- At the bottom of Figure 3 and Figure 4, did the authors order the random search trials by performance? Because the performance over the iterations seems to follow a power law. Given that it is random search, I would expect some flat regions given by hyperparameter configurations that are not optimal. Based on the figures it seems that the performance is constantly improving which is very surprising. The curve looks like a curve that is generated from training a model.

- Throughout the manuscript, the authors mention that they use 1024 iterations for each considered model/search space combination, however, on the plots the number of iterations is up to 100 for Figure 3 and up to 70 for Figure 4. Do the authors consider the number of repetitions too, how exactly is the number 1024 devised? How exactly are the 48 subsamples collected, part of the beginning of the "tuning curve" or randomly from the full data?

- **Line 502, "however, random search remains a strong baseline, with variants near state-of-the-art (Li et al.,2018;2020)."**

   The authors do not accurately reflect the current state of the domain. HyperBand and ASHA are not the state-of-the-art in multi-fidelity hyperparameter optimization. There have been several advancements that combined the schedule of HyperBand with model-based surrogates[1][2] and more recently, the current state-of-the-art [3][4] approaches that use an adaptive schedule with model-based algorithms. I would urge the authors to incorporate the provided citations into the manuscript to provide accurate information about the current state-of-the-art in multi-fidelity optimization.

- While the authors advocate the use of their proposed work with deep learning, the deep learning tasks are expensive and achieving $x$ trials is computationally demanding. In these scenarios, practitioners tend to use multi-fidelity based methods that are model-based. Random search is not a very promising algorithm.

- How many data points (HPO trials) are needed for the provided distribution to accurately reflect the tuning curve?

[1] Falkner, S., Klein, A., & Hutter, F. (2018, July). BOHB: Robust and efficient hyperparameter optimization at scale. In International conference on machine learning (pp. 1437-1446). PMLR.

[2] Awad, N., Mallik, N., & Hutter, F. DEHB: Evolutionary Hyberband for Scalable, Robust and Efficient Hyperparameter Optimization.

[3] Wistuba, M., Kadra, A., & Grabocka, J. (2022). Supervising the multi-fidelity race of hyperparameter configurations. Advances in Neural Information Processing Systems, 35, 13470-13484.

[4] Kadra, A., Janowski, M., Wistuba, M., & Grabocka, J. (2024). Scaling laws for hyperparameter optimization. Advances in Neural Information Processing Systems, 36.

---

> ### Comment · Reviewer_TdXJ · 2024-11-27
> **Rebuttal Reply**
>
> I have read all the other reviews and noticed that similar concerns to mine have been shared. Based on which I will keep my score and recommend for rejection.

---

> > ### Author Response · Authors · 2024-11-27
> >
> > Before finalizing your decision, please note the rebuttal period has not yet ended and allow us the opportunity to provide our response.

---

> ### Author Response · Authors · 2024-11-27
>
> Thank you for the references on state-of-the-art hyperparameter tuning, we have incorporated them into our related work. As clarified in our general response: hyperparameter tuning algorithms are not the subject of our work; nonetheless, some readers might find those references interesting. We should also clarify: we never claimed HyperBand or ASHA to be state-of-the-art, rather we intended that they are strong algorithms which are still compared to it (for example, in your reference: [4]). To make this unambiguous, we have implemented your suggestion and revised the language from “near state-of-the-art” to “obtaining high performance”.
>
> Before addressing your other concerns, we emphasize again: our work does not propose random search as an alternative to state-of-the-art and it does not seek the best hyperparameter tuning algorithm; rather, our work develops better tools for the design and analysis of deep learning experiments, and offers a better understanding of random search itself—including its limitations. It would be interesting to develop similar tools for other hyperparameter tuning algorithms; however, that is beyond the scope of this work.
>
> To address your individual concerns:
>
> *No baselines are considered*: The main task on which we evaluate is estimating and extrapolating confidence bands for tuning curves. We compare to Lourie et al. (2024) which, at present, offers the only confidence bands for tuning curves. We do not compare random search to other hyperparameter tuning algorithms as this is both out-of-scope and unnecessary&mdash;many such comparisons already exist in the literature.
>
> *Limited experiments*: As you point out, we evaluate on “diverse deep learning models” spanning both vision and language. Nevertheless, empirical claims can always be bolstered by more experiments. Accordingly, we have added new experiments with AlexNet [1] and ConvNext [2] in Appendix D. With these additions, we cover over a decade of advancements in architecture, three models from vision, two models from language, and both pretraining and supervised fine-tuning. Combined with our theoretical proofs, this offers substantial evidence for our claims.
>
> *Related work is outdated*: As discussed above, we have addressed your concerns about the related work and updated it with the references you provided.
>
> [1]: A. Krizhevsky, I. Sutskever, G. E. Hinton. (2012). ImageNet Classification with Deep Convolutional Neural Networks. Advances in Neural Information Processing Systems, 25.
>
> [2]: Z. Liu, H. Mao, C. -Y. Wu, C. Feichtenhofer, T. Darrell and S. Xie. (2022). A ConvNet for the 2020s. IEEE/CVF Conference on Computer Vision and Pattern Recognition (CVPR), pp. 11966-11976.

---

> ### Author Response · Authors · 2024-11-27
>
> Separate from our rebuttal above, we are happy to respond to your questions:
>
> *The region of relevant hyperparameters converges about the optimum*: You disagree that the “region of relevant hyperparameters” converges about the optimum because random search does not adapt. This objection comes from a misunderstanding of what was meant by relevant hyperparameters. In the sentence before the quote (line 180), we defined it as: “the [hyperparameters] better than the best you have seen so far”. Thus, the region of relevant hyperparameters is not a property of the algorithm at all, but how far the search has progressed. Under general conditions, the set of hyperparameters better than the current best does indeed converge about the optimum. We have added a formal proof of this (Proposition E.1). Based on your feedback, we have revised the language to be more specific and included a formal proof of our limit theorem in Appendix E.
>
> *“... being close to the optimum, would require a very very large number of trials in a continuous search space of D dimensions”*: This statement is mostly true, except the speed of convergence depends not on the dimension of the space but the *effective* dimension of the loss surface. In fact, our theory shows this and quantifies exactly how this dimension affects random search. In all five practical scenarios we considered, this dimension was low: 1 or 2.
>
> *“The work defines the asymptotic regime as the hyperparameters that we care about the most, those close to the optimum (Line 81).”*: Thank you for bringing this up, our language here was too imprecise. We define the asymptotic regime as the point where the asymptotics determine the behavior of random search. It is a bias-variance trade-off between how good the Taylor approximation is and how many data points you have to fit the distribution. We replaced this description with a more specific one (line 182).
>
> *“... did the authors order the random search trials by performance? … The curve looks like a curve that is generated from training a model”*: As explained in Section 2.1: Formalizing Random Search, we estimate the *median* tuning curve. While individual runs of random search display flat regions, the median of all such runs will be smooth because the probability of having found a good configuration increases with each iteration. The curves are not from training a model, and their construction is described in detail in Section 3. The median tuning curve is a statistical quantity, so we use a large sample of 1,024 iterations to estimate it. We only visualize the first 70 to 100 iterations to better show the curves’ structure, since they are essentially flat past that point.
>
> *“... practitioners tend to use multi-fidelity based methods that are model-based…”*: As discussed in our general response, random search (along with grid search) remains one of the most common hyperparameter tuning methods in practice for deep learning research. In the Llama 3.1 report from this year with over 200 core contributors, these are the only two hyperparameter tuning methods mentioned.
>
> *“How many data points (HPO trials) are needed… to accurately reflect the tuning curve”*: The answer to this question is necessarily subjective since it depends on the desired level of accuracy, but to get a sense of it we show fits obtained using 48 data points in Figure 6.

---

> > ### Comment · Reviewer_TdXJ · 2024-11-27
> > **Note**
> >
> > My apologies, I posted with the idea that the discussion period ended. However, it appears this year there exists a new design where the authors have one more day to reply. I will read your reply in detail and I will incorporate it in my judgement.

---

> > > ### Author Response · Authors · 2024-11-28
> > >
> > > Thank you very much! We greatly appreciate your additional consideration.

---

### Author Response · Authors · 2024-11-27

The title of our work lays out its core contribution: “An Asymptotic Theory of Random Search”. To deliver on this promise, we prove *a novel limit theorem*: the best scores from random search converge to the noisy quadratic distribution. This *new family of probability distributions* does not exist in the literature&mdash;we introduce it, derive its formulas, prove its properties, and implement its computational details. To enable widespread use, we make it *publicly available in our library*. Besides derivations and theoretical proof, we empirically validate our framework, demonstrating that the asymptotic distribution matches the ground truth across *five architectures spanning both vision and language*. We not only show our theory’s predictions match experiments, but verify its assumptions as well. Specifically, we show the noise from random seeds is normal and homoskedastic.

A number of reviewers expressed a shared misunderstanding: we do *not* propose random search as an alternative to state-of-the-art hyperparameter tuning methods. We do not study how to find the best hyperparameters at all. Rather, we aim to develop *better tools for the design and analysis of deep learning experiments*. As a secondary contribution, we also seek a better understanding of random search&mdash;both its strengths and its limitations.

Several reviewers point out that random search is not state-of-the-art. On this point, we completely agree: it is not state-of-the-art; however, it *is* practically significant. Along with grid search, random search remains one of the most common methods for hyperparameter tuning in typical deep learning experiments. For example, the Llama 3.1 report [1]—a substantial effort with over 200 core contributors and extensive experiments—mentions only two hyperparameter tuning methods: grid search (Section 7.5.2) and random search (Section 4.3.2). Thus, a theory of random search has considerable implications for deep learning practice. At the same time, random search has notable limitations; indeed, our theory clarifies them. It identifies the main determinants of random search’s performance (e.g., the effective number of hyperparameters), and *quantifies* exactly how they affect its progress. Sequential model based optimization can overcome these limitations; it offers a powerful tool and a fascinating area of research, and we hope future work builds off of our findings to analyze this more complex case.

A final concern raised by several reviewers was practical impact. While understanding random search has its own merits, our primary aim was to improve deep learning practice. We hoped the theory would provide statistical tools for use in deep learning experiments; indeed, it has. In Section 4.3, we demonstrate how to extrapolate confidence bands for model performance as a function of tuning effort. These bands can be used, for example, to compare optimizers where robustness to hyperparameters is essential. Other applications include estimating confidence intervals for the best hyperparameters’ performance, or determining the effective number of hyperparameters.

We provide both theoretical and empirical evidence for our claims. For ease of presentation, we keep theoretical discussion informal in the main text. To ensure mathematical rigor, we have added Appendix E, containing formal proofs. Besides proving the limit, we test our assumptions and demonstrate convergence empirically. Though the theory is asymptotic, in our experiments it characterizes the behavior of random search after 1-4 iterations.

All reviewers agree that our experiments support the theory, though some expressed concern that fast convergence might not hold for other architectures. This could happen if convergence is due to the architectures we considered being particularly easy to tune, or if our search spaces were unusually well-suited for them. In response, we have used *the same search space* from our ResNet18 experiments with AlexNet (an old, less advanced architecture) and ConvNext (a new, more advanced one). These architectures span a decade of deep learning advancements, and because we use the same search space across them, it can not be tailored to each. This setting reconfirms our results: the theory characterizes random search in the first 2-4 iterations. So, though surprising, the asymptotic regime really does describe random search after just a few iterations.

As a tool for better experiments, as a framework for understanding random search, and as a foundation for future analyses, our theory offers many benefits for those who deal with hyperparameters in their research.

[1]: Llama Team, AI @ Meta. The Llama 3 Herd of Models. (2024). https://arxiv.org/abs/2407.21783

---

### Meta-Review · Area_Chair_zKSn · 2024-12-22

**Metareview:**

In this paper, the authors establish a more formal understanding of random search as a method for tuning hyper-parameters of deep learning models, showing that it converges to a "noisy quadratic" distribution.  There are well established methodologies for performing experimental design for hyper-parameter tuning when experiments are noisy.  However, grid search and random search remain popular - presumably due to their simplicity and robustness to noise, non-stationarity, etc.  Therefore, having a better understanding of random search seems useful.  The reviewers found the new theoretical view novel, and the paper clear and insightful.  However, the reviewers all voted to reject the paper (3, 3, 5, 5).  Multiple reviewers seemed to question whether the proposed theory was relevant to practice - i.e. that the "asymptotic regime" was a reasonable assumption in practice.  Also that if the assumptions held, i.e. if the underlying function was smooth and well-approximated by a quadratic, then other tools would be more appropriate than random search.  Other reviewers also asked for comparison to other hyper parameter tuning methods.  (Note, the authors argue that comparing different tools was not a claim or objective of the paper - rather than understanding random search).  Reviewers also asked for theoretical justification of the noisy quadratic assumption, and more theoretical justification in general.  The authors seem to have added this in the response, but in the appendix. Finally, although the reviewers seemed to believe that the experiments supported the theory, they had concerns about whether the results would generalize across architectures, and questioned whether the search spaces were unusually well suited to random search.

While the paper seems insightful and relevant to how hyper-parameter tuning is done in practice, the reviews in sum seem to suggest that the paper is not quite ready for publication.  The scores would place the paper well below the threshold for acceptance.  Therefore, the recommendation is to reject.  However, there seems to be a good start here.  Given the reviews, it seems like it would be useful to establish, given the theory, when random search might be more appropriate than a model-based approach (high parallelism? lots of noise and non-stationarity?) and then provide some insight on best practices?  Hopefully the reviews will be useful to strengthen the paper for a future submission.

**Additional Comments On Reviewer Discussion:**

There was some discussion between the reviewers and authors.  In particular, the authors seemed to feel that the reviewers were missing the point of the paper - i.e. to provide some theoretical insight into random search rather than make claims about its performance compared to other methods.  In my view, if this was a consensus takeaway from the reviewers, then the narrative of the paper needs to be changed.

There were multiple questions about whether the smoothness assumption and quadratic assumption were valid?  The authors provided some theory in the response which seemed to convince one reviewer to raise their score (after they dropped it after reading other reviews).  However, the reviewers all kept their scores below the accept threshold.

There were also concerns about related work in that the state-of-the-art methods for hyper-parameter tuning weren't included in discussion.  The authors agreed to include these in the paper, but noted that they were not aiming to compare methods.

---

### Decision · Program_Chairs · 2025-01-22

Reject